# Counterfactual Fairness with Partially Known Causal Graph

**Aoqi Zuo**
The University of Melbourne
azuo@student.unimelb.edu.au

**Susan Wei**
The University of Melbourne
susan.wei@unimelb.edu.au

**Tongliang Liu**
The University of Sydney
tongliang.liu@sydney.edu.au

**Bo Han**
Hong Kong Baptist University
bhanml@comp.hkbu.edu.hk

**Kun Zhang**
Carnegie Mellon University & MBZUAI
kunz1@cmu.edu

**Mingming Gong**[†]
The University of Melbourne
mingming.gong@unimelb.edu.au

## Abstract

Fair machine learning aims to avoid treating individuals or sub-populations unfavourably based on *sensitive attributes*, such as gender and race. Those methods in fair machine learning that are built on causal inference ascertain discrimination and bias through causal effects. Though causality-based fair learning is attracting increasing attention, current methods assume the true causal graph is fully known. This paper proposes a general method to achieve the notion of counterfactual fairness when the true causal graph is unknown. To select features that lead to counterfactual fairness, we derive the conditions and algorithms to identify ancestral relations between variables on a *Partially Directed Acyclic Graph (PDAG)*, specifically, a class of causal DAGs that can be learned from observational data combined with domain knowledge. Interestingly, we find that counterfactual fairness can be achieved as if the true causal graph were fully known, when specific background knowledge is provided: the sensitive attributes do not have ancestors in the causal graph. Results on both simulated and real-world datasets demonstrate the effectiveness of our method.

## 1 Introduction

With the widespread application of machine learning in various fields (e.g., hiring decisions [23], recidivism predictions [1, 11], and finance [26, 55]), the ethical and social impact of machine learning is receiving increasing attention. In particular, machine learning algorithms are sensitive to the bias in training data, which may render their decisions discriminatory against individual or sub-population group with respect to *sensitive attributes*, e.g., gender and race. For example, bias against African-Americans was found with COMPAS, a decision support tool used by U.S. courts to assess the likelihood of a defendant becoming a recidivist [12].

To achieve fair machine learning, a large body of methods have been proposed to mitigate bias according to different fairness measures. These methods can be roughly categorized into two groups. The first group focuses on devising statistical fairness notions, which typically indicate the statistical

---

[†]Corresponding author

discrepancy between individuals or sub-populations, e.g., statistical parity [14], equalized odds [22], and predictive parity [8]. Built on the causal inference framework [40], the second group treats the presence of causal effect of the sensitive attribute on the decision as discrimination [4, 19, 25, 27–29, 37, 48, 49, 58, 60, 64, 65, 67–69].

Among the causal fairness works, counterfactual fairness [29], which considers the causal effect of sensitive attributes on the individual level, has received much attention. Given the true causal graph, Kusner et al. [29] provide the conditions and algorithms for achieving counterfactual fairness in the constructed predictive model. Wu et al. [60] and Chiappa [4] extend counterfactual fairness by considering path-specific effects. When the counterfactual effect is unidentifiable, counterfactual fairness can be approximately achieved by lower and upper bounds of counterfactual effects [59].

Existing counterfactual fairness methods assume the availability of a causal directed acyclic graph (DAG) [39], which encodes causal relationships between variables. However, in many real-world scenarios, the causal DAG is often unknown due to insufficient understanding of the system under investigation. An obvious path forward is to infer the causal DAG from observational data using causal discovery methods [6, 9, 24, 43, 44, 51–53, 66]. Unfortunately, without strong assumptions on the data generating process, such as linearity [51] and additive noise [24, 44], one cannot uniquely recover the underlying true causal graph from observational data alone. In the general case, causal discovery methods could output a Markov equivalence class of DAGs that encode the same set of conditional independencies from data, which can be represented by a completely partially directed acyclic graph (CPDAG) [6, 54]. With additional background knowledge, we can discern more causal directions, which can be represented by a maximally partially directed acyclic graph (MPDAG) [34], but we still cannot obtain a unique causal DAG.

Following this augment, we have a natural question to answer: can we learn counterfactual fairness with a partially known causal graph represented by MPDAG?[2] In a causal DAG, if a variable $S$ has a causal path to another variable $T$, then $S$ is an ancestor of $T$ and $T$ is a descendant of $S$. Counterfactual fairness deems the prediction to be counterfactually fair if it is a function of the non-descendants of sensitive attributes [29], which are straightforward to identify in DAGs. However, in an MPDAG, with respect to a variable $S$, a variable $T$ can be either

- a *definite descendant* of $S$ if $T$ is a descendant of $S$ in every equivalent DAG,
- a *definite non-descendant* of $S$ if $T$ is a non-descendant of $S$ in every equivalent DAG,
- a *possible descendant* of $S$ if $T$ is neither a definite descendant nor a definite non-descendant of $S$.

To achieve counterfactual fairness in MPDAGs, we need to select the definite non-descendants and some possible descendants of the sensitive attributes that could be non-descendants in the true DAG to make prediction. This comes to the core challenge: the identifiability of the ancestral relation between two distinct variables in an MPDAG. We refer the interested readers to a summary of existing identifiability results in Appendix G.

In this paper, we assume no selection bias and presence of confounders because the causal discovery algorithms themselves will not work well in such challenging scenarios. Under this assumption, which can also be found in the most related work [4, 7, 59, 68], but removing the assumption on a fully directed causal DAG, we make the following main contributions towards achieving counterfactual fairness on MPDAGs:

- We provide a sufficient and necessary graphical criterion (Theorem 4.5) to check whether a variable is a definite descendant of another variable on an MPDAG;
- Based on the proposed criterion, we give an efficient algorithm (Algorithm 2) to identify ancestral relations between any two variables on an MPDAG;
- We propose the first approach for achieving counterfactual fairness on partially known causal graphs, specifically MPDAGs;
- We find that on an MPDAG, counterfactual fairness can be achieved as if the true causal graph is fully known with the assumption that the sensitive attributes can not have ancestors in the causal graph.

---

[2]CPDAG is a special case of MPDAG without background knowledge, so we deal with MPDAG generally.

## 2 Background

In this section, we first review the structural causal model, causal graph and counterfactual inference. Then we introduce counterfactual fairness criterion and its intuitive implications.

### 2.1 Structural causal model, causal graph, and counterfactual inference

Structural causal model (SCM) [40] is a framework to model causal relations between variables. It is defined by a triple $(U, V, F)$, where $V$ are observable endogenous variables and $U$ are unobserved exogenous variables that cannot be caused by any variable in $V$. $F$ is a set of functions $f_1, ..., f_n$, one for each $V_i \in V$ expressing how $V_i$ is dependant on its direct causes: $V_i = f_i(pa_i, U_i)$, where $pa_i$ is the observed direct causes of $V_i$ and $U_i$ is the set of unobserved direct causes of $V_i$. The exogenous $U_i$s are required to be jointly independent. The set of equations $F$ induces a causal graph $\mathcal{D}$ over the variables, usually in the form of a directed acyclic graph (DAG), where the directed causes of $V_i$ represents its parent set in the causal graph.

Based on SCM, one can perform counterfactual inference to answer the problems in the counterfactual world. For example, consider in a fairness context, $A$, $Y$ and $\mathcal{X}$ represent the sensitive attributes, outcome of interest, and other observable attributes, respectively. For an individual $U = u$ with $A = a, Y = y$ and $\mathcal{X} = \mathbf{x}$, a common counterfactual query is: "For this individual $u$, what would the value of $Y$ have been had $A$ taken value $a'$". The solution, denoted as $Y_{A \leftarrow a'}(u)$ can be obtained by three steps in the deterministic case: Abduction, Action and Prediction [40, Chapter 7.1]. In probabilistic counterfactual inference, the procedure can be modified to estimate the posterior of $u$ and the distribution of $Y_{A \leftarrow a'}(u)$.

**DAGs, PDAGs and CPDAGs.** In a directed acyclic graph (DAG), all edges are directed and there is no directed cycle in the graph; when some edges are undirected, we say it is a partially directed graph (PDAG). A DAG encodes a set of conditional independence relations based on the notion *d-seperation* [38]. Multiple DAGs are Markov equivalent if they encode the same set of conditional independence relations. A *Markov equivalence class* of a DAG $\mathcal{D}$ can be uniquely represented by a completed partially directed acyclic graph (CPDAG) $\mathcal{G}^*$, denoted by $[\mathcal{G}^*]$.

**MPDAGs.** The CPDAGs with background knowledge constraint is known as maximally oriented PDAGs (MPDAGs) [34], which can be obtained by applying Meek's rules R1, R2, R3 and R4 in [34]. The Algorithm 1 in [42] can be used to construct the MPDAG $\mathcal{G}$ from the CPDAG $\mathcal{G}^*$ and backgroud knowledge $\mathcal{B}$, where the background knowledge $\mathcal{B}$ is assumed to be the *direct causal information* in the form $X \rightarrow Y$, meaning that $X$ is a direct cause of $Y$. The subset of Markov equivalent DAGs consistent with the background knowledge $\mathcal{B}$ can be uniquely represented by an MPDAG $\mathcal{G}$, denoted by $[\mathcal{G}]$. Both a DAG and a CPDAG can be regarded as special cases of an MPDAG when the background knowledge is completely known and not known, respectively.

### 2.2 Counterfactual fairness

Counterfactual fairness [29] is a fairness criterion based on SCM [40]. Let $A$, $Y$ and $\mathcal{X}$ represent the sensitive attributes, outcome of interest and other observable attributes, respectively and the prediction of $Y$ is denoted by $\hat{Y}$. For an individual with $\mathcal{X} = \mathbf{x}$ and $A = a$, we say the prediction $\hat{Y}$ is counterfactually fair if it would have been the same had $A$ been $a'$ in the counterfactual world as in the real world that $A$ is $a$.

**Definition 2.1** (Counterfactual fairness). *[29, Definition 5] We say the prediction $\hat{Y}$ is counterfactually fair if under any context $\mathcal{X} = \mathbf{x}$ and $A = a$,*

$$P(\hat{Y}_{A \leftarrow a}(U) = y | \mathcal{X} = \mathbf{x}, A = a) = P(\hat{Y}_{A \leftarrow a'}(U) = y | \mathcal{X} = \mathbf{x}, A = a),$$

*for all $y$ and any value $a'$ attainable by A.*

The definition of counterfactual fairness immediately suggests the following approach in Lemma 2.2 to design a counterfactually fair model.

**Lemma 2.2.** *[29, Lemma 1] Let $\mathcal{G}$ be the causal graph of the given model $(U, V, F)$. Then $\hat{Y}$ will be counterfactually fair if it is a function of the non-descendants of A.*

# 3 Problem formulation

In this section, we introduce the task of achieving counterfactual fairness given PDAGs, especially MPDAGs that can be learned from observational data using causal discovery algorithms [5, 9, 53].

Lemma 2.2 in Section 2.2 implies learning a counterfactually fair prediction can be framed as selecting the non-descendants of $A$ to predict $Y$. If a causal DAG is used to encode the causal relations of all attributes, finding all non-descendants of $A$ is straightforward. However, as mentioned in Section 1, given observational data and optional background knowledge about direct causal information, we can only learn a CPDAG or an MPDAG $\mathcal{G}$, instead of the true DAG $\mathcal{D} \in [\mathcal{G}]$. Unfortunately, not all ancestral relations between $A$ and attributes in $\mathcal{X}$ are identifiable in a CPDAG or MPDAG.

Therefore, to achieve counterfactually fair prediction, we have two problems to solve:

- Identify the type of ancestral relations of any other attributes with $A$ in $\mathcal{G}$, *i.e.*, identifying the definite non-descendants, definite descendants, and possible descendants of $A$ in an MPDAG (Section 4);
- Build a counterfactually fair model based on the identified ancestral relations (Section 5).

# 4 Identifiability of ancestral relations in MPDAGs

In this section, we give a sufficient and necessary graphical criterion on identifying the definite ancestral relations between two distinct vertices in an MPDAG. We also provide an efficient algorithm for implementing the proposed criterion. We denote the parents, children, siblings and adjacencies of the node $W$ in a graph $\mathcal{G}$ as $pa(W, \mathcal{G})$, $ch(W, \mathcal{G})$, $sib(W, \mathcal{G})$ and $adj(W, \mathcal{G})$, respectively. A *chord* of a path in $\mathcal{G}$ is any edge joining two non-consecutive vertices on the path. A path without any chord is called *chordless path*. In a graph $\mathcal{G} = (V, E)$, where $V$ and $E$ represent the node set and edge set in $\mathcal{G}$, the *induced subgraph* of $\mathcal{G}$ over $V' \subset V$ is the graph with vertex $V'$ and edges between vertices in $V'$, that is $E' \subset E$. A graph is *complete* if any two distinct vertices are adjacent.

## 4.1 Graphical criterion on identifying ancestral relations in MPDAGs

We first introduce the term *b-possibly causal path* [42] in MPDAGs where the prefix *b-* stands for background.

**Definition 4.1** (b-possibly causal path, b-non-causal path). *[42, Definition 3.1] Suppose $p = \langle S = V_0, ..., V_k = T \rangle$ is a path from $S$ to $T$ in an MPDAG $\mathcal{G}$, $p$ is b-possibly causal in $\mathcal{G}$ if and only if no edge $V_i \leftarrow V_j$, $0 \le i \le j \le k$ is in $\mathcal{G}$, including the edge not on $p$. Otherwise, $p$ is a b-non-causal path in $\mathcal{G}$.*

Perković et al. [42] state that $T$ is a definite non-descendant of $S$ in an MPDAG $\mathcal{G}$ if and only if there is no b-possibly causal path from $S$ to $T$ in $\mathcal{G}$. In this section, to identify whether $T$ is a definite descendant of $S$ in an MPDAG, we provide a sufficient and necessary condition.

We introduce the term *critical set* [15] in CPDAGs as follows.

**Definition 4.2** (Critical Set). *[15, Definition 2] Let $\mathcal{G}$ be an CPDAG, $S$ and $T$ be two distinct vertices in $\mathcal{G}$. The critical set of $S$ with respect to $T$ in $\mathcal{G}$ consists of all adjacent vertices of $S$ lying on at least one chordless possibly causal path from $S$ to $T$.*

Definition 4.2 can be extended to MPDAGs directly, based on which, we provide a sufficient and necessary graphical condition in Lemma 4.3 for identifying the definite ancestral relation between two distinct vertices in an MPDAG.

**Lemma 4.3.** *Let $\mathcal{G}$ be an MPDAG and $S$, $T$ be two distinct vertices in $\mathcal{G}$, then $T$ is a definite descendant of $S$ in $\mathcal{G}$ if and only if the critical set of $S$ with respect to $T$ always contains a child of $S$ in every DAG $\mathcal{D} \in [\mathcal{G}]$.*

The proof of Lemma 4.3 is in Appendix B.1. Note that we have to enumerate all Markov equivalent DAGs for checking the condition given in Lemma 4.3. To resolve this problem, we provide a condition in Lemma 4.4 to check the graphical characteristic of the corresponding critical set in the MPDAG directly. The graphical criterion in Lemma 4.4 has been proved by [16, Lemma 2] and [3, Lemma 1] for CPDAGs. We extend it to general MPDAGs here.

**Lemma 4.4.** *Let $\mathcal{G}$ be an MPDAG and $S$, $T$ be two distinct vertices in $\mathcal{G}$. Denote by $\mathbf{C}$ the critical set of $S$ with respect to $T$ in $\mathcal{G}$, then $\mathbf{C} \cap ch(S, \mathcal{D}) = \varnothing$ for some DAG $\mathcal{D} \in [\mathcal{G}]$, if and only if $\mathbf{C} = \varnothing$, or $\mathbf{C}$ induces a complete subgraph of $\mathcal{G}$ but $\mathbf{C} \cap ch(S, \mathcal{G}) = \varnothing$.*

The proof of Lemma 4.4 is in Appendix B.2. Building on Lemma 4.3 and Lemma 4.4, we arrived at the desired sufficient and necessary graphical criterion in Theorem 4.5.

**Theorem 4.5.** *Let $S$ and $T$ be two distinct vertices in an MPDAG $\mathcal{G}$, and $\mathbf{C}$ be the critical set of $S$ with respect to $T$ in $\mathcal{G}$. Then $T$ is a definite descendant of $S$ if and only if either $S$ has a definite arrow into $\mathbf{C}$, that is $\mathbf{C} \cap ch(S, \mathcal{G}) \neq \varnothing$, or $S$ does not have a definite arrow into $\mathbf{C}$ but $\mathbf{C}$ is non-empty and induces an incomplete subgraph of $\mathcal{G}$.*

With Theorem 4.5, we can identify whether $T$ is a definite descendant of $S$ in an MPDAG by finding all chordless b-possibly causal path from $S$ to $T$ and checking graphical characteristic of the corresponding critical set. We provide an example to illustrate Theorem 4.5 in Appendix C.

## 4.2 Algorithms

Here we provide an efficient algorithm to identify the ancestral relation between any two distinct vertices in an MPDAG based on the theoretical results in Theorem 4.5. It is straightforward to identify the non-ancestral relation by checking if there exists a b-possibly causal path from the source to the target. However, discriminating between a definite ancestral relation and a possible ancestral relation in an MPDAG by Theorem 4.5 requires finding the critical set.

According to the definition of the critical set, we need to find all chordless possibly causal paths from the source variable to the target variable first. However, it is cumbersome to check whether a path is chordless, since it involves considering many edges not on the path. In Proposition 4.7 we propose a more efficient way to find critical sets by leveraging the relation between chordless path and definite status path in Lemma 4.6. We first provide the notion of *collider* and *definite status path*.

**Colliders and Definite Status paths.** In a path $p = \langle S = V_0, ..., V_k = T \rangle$, if $V_{i-1} \to V_i$ and $V_i \leftarrow V_{i+1}$, then we say $V_i$ is a *collider* on $p$. If $V_{i-1}$ and $V_{i+1}$ are not adjacent, then the triple $\langle V_{i-1}, V_i, V_{i+1} \rangle$ is called a *v-structure collided* on $V_i$, and $V_i$ can be called *unshielded collider*. A node $V_i$ is a *definite non-collider* on a path $p$ if there is at least one edge out of $V_i$ on $p$ or $V_{i-1} - V_i - V_{i+1}$ is a subpath of $p$ and $V_{i-1}$ is not adjacent with $V_{i+1}$. A node is of *definite status* on a path if it is a collider, a definite non-collider or an endpoint on the path. A path $p$ is of definite status if every node on $p$ is of definite status.

**Lemma 4.6.** *Let $S$ and $T$ be two distinct vertices in an MPDAG $\mathcal{G}$. If $p$ is a chordless path from $S$ to $T$ in $\mathcal{G}$, then $p$ is of definite status.*

**Proposition 4.7.** *Let $\mathcal{G}$ be an MPDAG, $S$ and $T$ be two distinct vertices in $\mathcal{G}$. Denote by $\mathbf{C}_{ST}$ the critical set of $S$ with respect to $T$ in $\mathcal{G}$, then $\mathbf{C}_{ST} = \mathbf{F}_{ST}$, where $\mathbf{F}_{ST}$ denotes all adjacent vertices of $S$ lying on at least one b-possibly causal path of definite status from $S$ to $T$ in $\mathcal{G}$ that there is no chord with $S$ as an endpoint.*

Proposition 4.7 implies that, instead of enumerating all chordless possibly causal path from the source vertex to the target vertex, we enumerate b-possibly causal paths of definite status, excluding paths that contain any chord where the source vertex is an end node. The nodes adjacent to the source vertex on these paths constitute the desired critical set.

The proof of Lemma 4.6 and Proposition 4.7 are in Appendix B.4 and Appendix B.5, respectively.

Following from Proposition 4.7, we develop Algorithm 1 showing how to efficiently find the critical set of $S$ w.r.t. $T$ in an MPDAG $\mathcal{G}$. Given an MPDAG $\mathcal{G}$, the source vertex $S$ and the target vertex $T$, the output of Algorithm 1 is the critical set of $S$ w.r.t. $T$ in $\mathcal{G}$. Using the breadth-first-search algorithm, Algorithm 1 searches b-possibly causal path of definite status without any chord ending in $S$ from $S$ to $T$. A detailed explanation of this algorithm is provided in Appendix D.

Finally, we present Algorithm 2 to identify the type of ancestral relation. We first find the critical set $\mathbf{C}$ of $S$ with respect to $T$. When $\mathbf{C} = \varnothing$, there is no b-possibly causal path from $S$ to $T$, so $T$ is a definite non-descendant of $S$. When $\mathbf{C} \neq \varnothing$, using Theorem 4.5, we can decide whether $T$ is a definite descendant or a possible descendant of $S$.

---

**Algorithm 1** Finding the critical set of $S$ with respect to $T$ in an MPDAG

---

1: **Input:** MPDAG $\mathcal{G}$, two distinct vertices $S$ and $T$ in $\mathcal{G}$.
2: **Output:** The critical set $\mathbf{C}$ of $S$ with respect to $T$ in $\mathcal{G}$.
3: Initialize $\mathbf{C} = \varnothing$, a waiting queue $\mathcal{Q} = [\,]$, and a set $\mathcal{H} = \varnothing$,
4: **for** $\alpha \in sib(S) \cup ch(S)$ **do**
5:     add $(\alpha, S, \alpha)$ to the end of $\mathcal{Q}$,
6: **while** $\mathcal{Q} \neq \varnothing$ **do**
7:     take the first element $(\alpha, \phi, \tau)$ out of $\mathcal{Q}$ and add it to $\mathcal{H}$;
8:     **if** $\tau = T$ **then**
9:         add $\alpha$ to $\mathbf{C}$, and remove from $\mathcal{S}$ all triples where the first element is $\alpha$;
10:     **else**
11:         **for** each node $\beta$ in $\mathcal{G}$ **do**
12:             **if** $\tau \to \beta$ or $\tau - \beta$ **then**
13:                 **if** $\tau \to \beta$ or $\phi$ is not adjacent with $\beta$ or $\tau$ is the endnode **then**
14:                     **if** $\beta$ and $S$ are not adjacent **then**
15:                         **if** $(\alpha, \tau, \beta) \notin \mathcal{H}$ and $(\alpha, \tau, \beta) \notin \mathcal{Q}$ **then**
16:                             add $(\alpha, \tau, \beta)$ to the end of $\mathcal{Q}$,
17: **return $\mathbf{C}$**

---

---

**Algorithm 2** Identify the type of ancestral relation of $S$ with respect to $T$ in an MPDAG

---

1: **Input:** MPDAG $\mathcal{G}$, two distinct variables $S$ and $T$ in $\mathcal{G}$.
2: **Output:** The type of ancestral relation between $S$ and $T$.
3: Find the critical set $\mathbf{C}$ of $S$ with respect to $T$ in $\mathcal{G}$ by Algorithm 1.
4: **if** $|\mathbf{C}| = 0$ **then**
5:     **return** $T$ is a definite non-descendant of $S$.
6: **if** $S$ has an arrow into $\mathbf{C}$ or $\mathbf{C}$ induces an incomplete subgraph of $\mathcal{G}$ **then**
7:     **return** $T$ is a definite descendant of $S$.
8: **return** $T$ is a possible descendant of $S$.

---

Since in Algorithm 1, the same triple like $(\alpha, \phi, \tau)$ can only be visited at most once, where $\alpha$ is a sibling or a child of $S$ in the MPDAG $\mathcal{G}$, $\tau$ is a node on a b-possibly causal path of definite status from $S$ to $T$ without any chord ending in $S$, and $\phi$ lies immediately before $\tau$ on such path. The complexity of Algorithm 1 in the worst case is $\mathcal{O}(|sib(S, \mathcal{G}) + ch(S, \mathcal{G})| * |E(\mathcal{G})|)$, where $|E(\mathcal{G})|$ is the number of edges in $\mathcal{G}$. Consequently, the computational complexity of Algorithm 2 is $\mathcal{O}(|sib(S, \mathcal{G}) + ch(S, \mathcal{G})| * |E(\mathcal{G})|)$.

## 5 Counterfactual fairness in MPDAGs

Now, we return to our problem of learning counterfactually fair models via selecting features from $\mathcal{X}$. We will consider two cases: 1) a general MPDAG and 2) an MPDAG learned with background knowledge that $A$ is a root node.

### 5.1 General case

We can identify the set of definite descendants, possible descendants and definite non-descendants of the sensitive attribute by simply applying Algorithm 2 to each pair of sensitive and any other attribute, which is concluded in Algorithm 4. The detailed description of Algorithm 4 is provided in Appendix E. As the computational complexity of Algorithm 2 in the worst case is $\mathcal{O}(|sib(S, \mathcal{G}) + ch(S, \mathcal{G})| * |E(\mathcal{G})|)$, the complexity of Algorithm 4 is directly $\mathcal{O}(|sib(S, \mathcal{G}) + ch(S, \mathcal{G})| * |E(\mathcal{G})| * |V(\mathcal{G})|)$, where $|E(\mathcal{G})|$ is the number of edges and $|V(\mathcal{G})|$ is the number of nodes in $\mathcal{G}$. We consider two feature selection methods. The first method (called Fair) only selects the definite non-descendants, which ensures counterfactual fairness. However, the number of definite non-descendants in an MPDAG might be too small, resulting in low prediction accuracy. Therefore, we also propose a second method (called FairRelax),which uses possible descendants of $A$ to increase the prediction accuracy at the cost of a violation of counterfactual fairness.

## 5.2 Under root node assumption

Kusner et al. [29, Section 3.2] mentioned the ancestral closure of sensitive attributes, meaning that typically we should expect the sensitive attribute set $A$ to be closed under ancestral relationships given by the causal graph. For instance, in the example that religion can be affected by the geographical place of origin, if *religion* is a sensitive attribute and *geographical place of origin* is a parent of *religion*, then it should also be in $A$. Therefore, $A$ will be the root node in most cases except some counterintuitive scenarios. Thus, we consider the following assumption, which is often true in real-world datasets.

**Assumption 5.1.** *The sensitive attribute can only be a root node in a causal MPDAG.*

For example, 'sex' cannot be caused by factors like 'education' and 'salary'. In an MPDAG $\mathcal{G}$, given Assumption 5.1, the ancestral relation between the sensitive attribute and any other attribute is fully identified, as shown in the following proposition.

**Proposition 5.2.** *In a MPDAG $\mathcal{G}$ with sensitive attribute $A$, if Assumption 5.1 holds, then any other attribute is either a definite descendant or definite non-descendant of $A$. Moreover, an arbitrary attribute $W$ is a definite descendant of $A$ if and only if there is a causal path from $A$ to $W$ in $\mathcal{G}$.*

The proof of Proposition 5.2 is in Appendix B.6. From Proposition 5.2, it is very interesting to see that fitting a model with the definite non-descendants of $A$ in $\mathcal{G}$ is exactly the same thing as fitting a model with the non-descendants of $A$ in the true DAG $\mathcal{D}$. Thus, the counterfactual fairness can be achieved as if the true causal DAG is fully known.

Under Assumption 5.1, since there is a causal path from $A$ to any definite descendant of $A$, we can directly identify whether a target attribute is a descendants of $A$ by checking if there is a causal path from $A$ to the target. To find all definite descendants in an MPDAG, we can use a breath first search algorithm with computational complexity $\mathcal{O}(|V| + |E|)$, where $|V|$ is the number of nodes and $|E|$ is the number of edges in $\mathcal{G}$. Then the remaining nodes are definite non-descendants of $A$ in $\mathcal{G}$.

# 6 Experiment

In this section, we illustrate our approach on a simulated and a real-world dataset by evaluating the prediction performance and fairness of our approach. The prediction performance is evaluated by root mean squared error (RMSE). The counterfactual unfairness can be measured by the discrepancy of the predictions in the real world and counterfactual world for each individual. In addition, unfairness can also be revealed by comparing the distributions of the predictions in two worlds, which coincide if the prediction is counterfactually fair.

**Baselines.** We consider three baselines: 1) `Full` is a standard model that uses all attributes, including the sensitive attributes to make predictions, 2) `Unaware` is a model that uses all attributes except the sensitive attributes to make predictions, and 3) `Oracle` is a model that makes predictions with all attributes that are non-descendants of the sensitive attribute given the groundtruth DAG. As mentioned in Section 5, our proposed methods include `FairRelax`, which makes predictions using all definite non-descendants and possible descendants of the sensitive attribute in an MPDAG, and `Fair`, which makes predictions using all definite non-descendants of the sensitive attribute in an MPDAG.

## 6.1 Synthetic data

The synthetic data is generated from linear structural equation models according to a random DAG. As the simulated DAG is known, we obtain the CPDAG from the true DAG without running the causal discovery algorithms.[3] We also add background knowledge to turn a CPDAG into its corresponding MDPAG.

We first randomly generate DAGs with $d$ nodes and $2d$ directed edges from the graphical model Erdős-Rényi (ER), where $d$ is chosen from $\{10, 20, 30, 40\}$. For each setting, we generate 100 DAGs. For each DAG $\mathcal{D}$, two nodes are randomly chosen as the outcome and the sensitive attribute, respectively. The sensitive attribute can have two or three values, drawn from a Binomial([0,1]) or

---

[3] Given a sufficiently large sample size, current causal discovery algorithms can recover the CPDAG with high accuracy on the simulated data [20].

Multinomial([0,1,2]) distribution separately. The weight, $\beta_{ij}$, of each directed edges $X_i \rightarrow X_j$ in the generated DAG, is drawn from a Uniform($[-2, -0.5] \cup [0.5, 2]$) distribution. The data are generated according to the following linear structural equation model:

$$X_i = \sum_{X_j \in pa(X_i)} \beta_{ij} X_j + \epsilon_i, i = 1, ..., n, \tag{1}$$

where $\epsilon_1, ..., \epsilon_n$ are independent $N(0, 1.5)$. Then we generate one sample with size 1000 for each DAG. The proportion of training and test data is splitted as $0.8 : 0.2$. Once the CPDAG $\mathcal{G}^*$ is obtained, where $\mathcal{D} \in [\mathcal{G}^*]$, we randomly generate the direct causal information $A \rightarrow B$ as the background knowledge from the edges where $A \rightarrow B$ is in DAG $\mathcal{D}$, while $A - B$ is in CPDAG $\mathcal{G}^*$. We show a randomly generated DAG $\mathcal{D}$, the corresponding CPDAG $\mathcal{G}^*$ and MPDAG $\mathcal{G}$ as in Figure 8, see Appendix F.1. For additional experiments based on more complicated structural equations and varying amount of possible background knowledge, please refer to Appendix F.4 and Appendix F.5. We also analyze the model robustness experimentally on causal discovery algorithms in Appendix F.6.

**Counterfactual fairness.** According to the predefined linear causal model, we first generate the counterfactual data given counterfactual sensitive attributes. For each individual, the noise of any counterfactual feature is the same as that in the observational data. In order to evaluate the counterfactual fairness of the baseline methods, we sample data from both the original and counterfactual data and fit them with all the models. Here, unfairness can be measured by the absolute difference of two predictions, $\hat{Y}_{A \leftarrow a}(u)$ and $\hat{Y}_{A \leftarrow a'}(u)$. The results for each model with different graph settings are shown in Table 1 and Figure 1a. Obviously, Oracle and Fair is counterfactually fair, since they do not use any feature that is causally dependant on the sensitive attribute. Full and Unaware have high counterfactual unfairness, while our FairRelax has very low counterfactual unfairness. Additionally, when the model is counterfactually fair, the distributions of the predictions in two worlds should lie on top of each other, as in the Oracle and Fair models. Although counterfactual unfairness is exhibited in the other three models, FairRelax is closer to strictly counterfactually fair methods. An exemplary density plot of the predictions of all the models in one original and counterfactual dataset is shown in Figure 2.

Table 1: Average unfairness and RMSE for synthetic datasets on held-out test set. For each graph setting, the unfairness gets decreasing from left to right and the RMSE gets increasing from left to right.

|  | Node | Edge | Full | Unaware | FairRelax | Oracle | Fair |
|---|---|---|---|---|---|---|---|
| Unfairness | 10 | 20 | $0.288 \pm 0.363$ | $0.200 \pm 0.322$ | $0.023 \pm 0.123$ | $0.000 \pm 0.000$ | $0.000 \pm 0.000$ |
| | 20 | 40 | $0.203 \pm 0.341$ | $0.165 \pm 0.312$ | $0.019 \pm 0.145$ | $0.000 \pm 0.000$ | $0.000 \pm 0.000$ |
| | 30 | 60 | $0.155 \pm 0.304$ | $0.143 \pm 0.312$ | $0.020 \pm 0.123$ | $0.000 \pm 0.000$ | $0.000 \pm 0.000$ |
| | 40 | 80 | $0.095 \pm 0.189$ | $0.075 \pm 0.182$ | $0.009 \pm 0.055$ | $0.000 \pm 0.000$ | $0.000 \pm 0.000$ |
| RMSE | 10 | 20 | $0.621 \pm 0.251$ | $0.637 \pm 0.261$ | $1.031 \pm 0.751$ | $1.065 \pm 0.751$ | $1.137 \pm 0.824$ |
| | 20 | 40 | $0.595 \pm 0.255$ | $0.599 \pm 0.253$ | $0.818 \pm 0.488$ | $0.847 \pm 0.55$ | $0.952 \pm 0.645$ |
| | 30 | 60 | $0.597 \pm 0.24$ | $0.601 \pm 0.242$ | $0.797 \pm 0.489$ | $0.849 \pm 0.644$ | $1.024 \pm 0.908$ |
| | 40 | 80 | $0.600 \pm 0.273$ | $0.601 \pm 0.272$ | $0.755 \pm 0.441$ | $0.766 \pm 0.452$ | $0.800 \pm 0.480$ |

**Accuracy.** For each graph setting, we report average RMSE achieved on 100 causal graphs by fitting a linear regression model for the baselines and our proposed models in Table 1 and Figure 1b. We can observe that, for each graph setting, the Full model obtains the lowest RMSE, which is not surprising as it uses all features. In addition, our FairRelax methods obtains better accuracy than the strictly counterfactually fair methods Fair and Oracle, which is consistent with the accuracy-fairness trade-off phenomenon. More discussion on accuracy-fairness trade-off is included in Appendix H.

## 6.2 Real data

The UCI Student Performance Data Set [10] regarding students performance in Mathematics is used in this experiment. The data attributes include student grade in secondary education, demographic, social, and school related features. It contains 395 students records with 32 attributes. We regard *sex* as the sensitive attribute in this dataset. Besides, we remove the first, second and final period grade, denoted by *G1,G2*, and *G3* in the dataset and generate the value of target attribute *Grade* as the average of *G1,G2*, and *G3*.

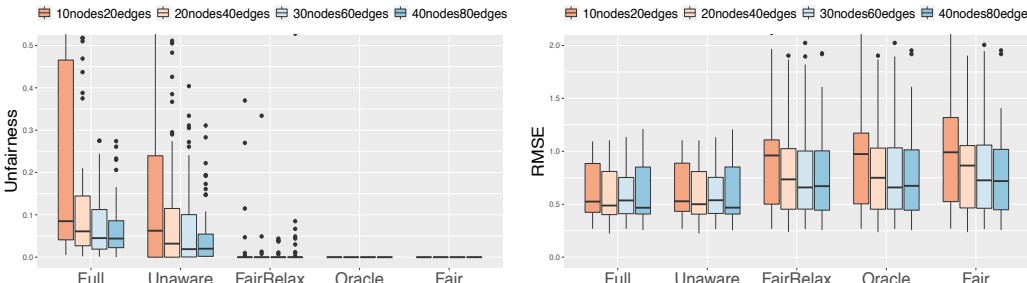

(a) Average unfairness for each model and graph setting. (b) Average RMSE for each model and graph setting.

Figure 1: Average unfairness and RMSE for synthetic datasets on held-out test set. For each graph setting, the unfairness gets decreasing from left to right, while RMSE has the opposite trend. The extreme unfairness and high RMSE is because almost all attributes are descendants of the sensitive attribute in around $10/100$ randomly generated graphs. This also explains why the standard deviation of unfairness for the model `Full`, `Unware`, `FairRelax` and RMSE for `FairRelax`, `Oracle` and `Fair` is that large in Table 1.

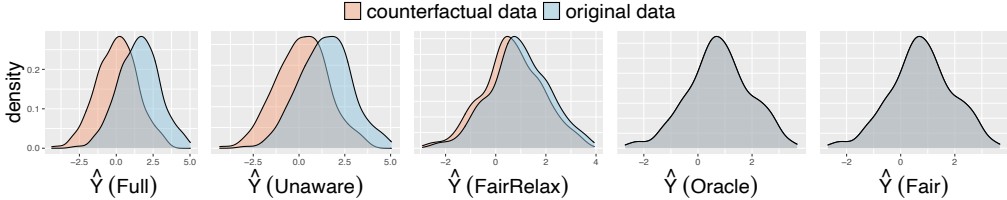

Figure 2: Density plot of the predicted $Y_{A\leftarrow a}(u)$ and $Y_{A\leftarrow a'}(u)$ in synthetic data.

In this section, we first learn the corresponding CPDAG from this dataset leveraging the GES structure learning algorithm [5], which is implemented by a general causal discovery software - TETRAD [45]. After uploading the preprocessed data, we can learn the CPDAG $\mathcal{G}^*$. The evolution of the CPDAG to MPDAG is shown in Appendix F.2.

Our experiments are carried out under the root node assumption on the MPDAG $\mathcal{G}$ in Figure 9c, as *sex* cannot be caused by other variables in the dataset. Due to the space limit, Figure 9c is provided in Appendix F.2. Thus, the ancestral relations can be fully identified according to our theoretical results in Section 5.2. Our algorithm will find all the non-descendants of the sensitive attribute. On this dataset, the definite descendants of *sex* can be identified as *{Walc, goout, Dalc, studytime}* and all the other nodes are definite non-descendants of *sex*.

The counterfactual fairness and accuracy are measured in almost the same way as in Section 6.1. The details on counterfactual data generation and model fitting can be referred to Appendix F.3. The results are reported in Table 2. Since under the root node assumption, there is no possible descendants of the sensitive attribute, the model `Fair` and `FairRelax` give the same RMSE result and both of them achieve counterfactual fairness at the cost of slight accuracy decrease. Instead, the model `Full` and `Unaware` are unfair and the `Full` model is more unfair than `Unaware`. Besides, the distribution of predictions for the original and counterfactual data for all models have the same trend as the synthetic data. Figure 3 is the corresponding density plot.

## 7    Conclusion and discussion

In this paper, we have developed a general approach to achieve counterfactual fairness when the true causal graph is unknown. In order to select features that lead to counterfactual fairness, we propose a sufficient and necessary condition and an efficient algorithm to identify the ancestral relations between any distinct vertices on an MPDAG, which may be applied to more applications.

Table 2: Average RMSE and unfairness for Student dataset on held-out test sets.

|  | Full | Unaware | FairRelax | Fair |
|---|---|---|---|---|
| Unfairness | $0.761 \pm 0.228$ | $0.250 \pm 0.085$ | $0.000 \pm 0.000$ | $0.000 \pm 0.000$ |
| RMSE | $3.49 \pm 0.292$ | $3.482 \pm 0.297$ | $3.509 \pm 0.356$ | $3.509 \pm 0.356$ |

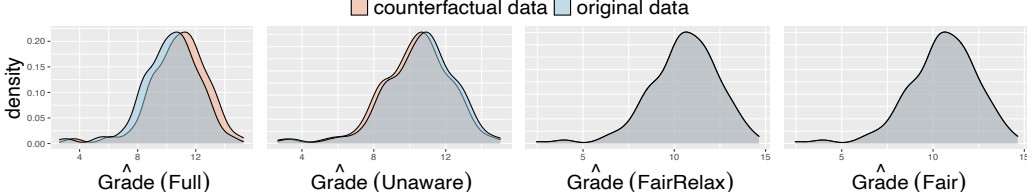

Figure 3: Density plot of the predicted $Grade_{sex \leftarrow a}(u)$ and $Grade_{sex \leftarrow a'}(u)$ for Student dataset.

Furthermore, an intriguing finding is that, under the assumption that the sensitive attribute can only be a root node in the graph, there is no possible descendant of the sensitive attribute, so that the fair features can be selected correctly whatever the true DAG is. Experiments on synthetic and real-world dataset show the effectiveness of our method.

One may claim that the fair prediction could be a function of the descendants of the sensitive attributes by balancing the observables, thus making the effect of the sensitive attribute canceled out. We agree with this opinion. However, it concerns the structural equations, which are in general unfalsifiable even if interventional data for all variables is available. In this paper, we do not address the assumptions on structural equations for achieving counterfactual fairness on MPDAGs. As a first step to obtaining a counterfactual fair predictor on an MPDAG, we focus on utilising the property of the causal graph (Level 1 in [29]) — making prediction with the definite non-descendants (and possible descendants) of the sensitive attribute. But our work can also be extended to the case where "cancel out" could happen on an MPDAG. One possible idea is to learn latent variables by specifying the structural equations (or more relaxed, conditional distributions). However, due to the fact that an MPDAG represents a set of DAGs with different conditional distributions components and thus enjoy different latent space, the intuitive way to enumerate all DAGs is unrealistic. Addressing such issue on an MPDAG is an interesting future direction.

Following prior work on establishing counterfactual fairness [4, 7, 59, 68], we assumed no selection bias and the presence of confounders throughout this paper. Yet, another fundamental assumption in these earlier studies in causal modelling is that the causal graph is known, which offers a new chance for the bias induced by misspecifying the causal DAG. Our work gives the first method to achieve counterfactual fairness without requiring the causal DAG to be specified. In the presence of selection bias and confounders on a causal DAG, a counterfactual fairness measure degenerates to demographic parity, which is discussed extensively by Fawkes et al. [17]. In this situation, it is considerably more difficult to provide a clear causal interpretation without specifying the causal DAG. This gives rise to an exciting topic to research in future work. Exploring causal discovery algorithms to find the ground-truth causal graph in the presence of selection bias and confounders and then achieving counterfactual fairness on partially ancestral graphs [46, 63], which would be significantly more difficult, is another possible future direction.

## 8   Acknowledgement

AZ was supported by Melbourne Research Scholarship from the University of Melbourne. SW was supported by ARC DE200101253. TL was partially supported by Australian Research Council Projects DP180103424, DE-190101473, IC-190100031, DP-220102121, and FT-220100318. BH was supported by NSFC Young Scientists Fund No. 62006202 and Guangdong Basic and Applied Basic Research Foundation No. 2022A1515011652. KZ was partially supported by the National Institutes of Health (NIH) under Contract R01HL159805, by the NSF-Convergence Accelerator Track-D award #2134901, by a grant from Apple Inc., and by a grant from KDDI Research Inc.. MG was supported by ARC DE210101624.

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
