# A Preliminaries

**Graph and Path.** Let $p = \langle S = V_0, ..., V_k = T \rangle$ be a path in a graph $\mathcal{G}$, $p$ is a *causal path* from $S$ to $T$ if $V_i \to V_{i+1}$ for all $0 \le i \le k - 1$. $p$ is a *possibly causal path* from $S$ to $T$ if no edge $V_i \leftarrow V_{i+1}$ is in $\mathcal{G}$. Otherwise, $p$ is a *non-causal path* in $\mathcal{G}$. A (causal, possibly causal, non-causal) cycle is a (causal, possibly causal, non-causal) path from a vertex to itself.

**Ancestral Relations.** If there is $S \to T$ in $\mathcal{G}$, we say $S$ is a parent of $T$ and $T$ is a child of $S$, denoted by $pa(T, \mathcal{G})$ and $ch(S, \mathcal{G})$, respectively. If there is a causal path from $S$ to $T$, then we say $S$ is an ancestor of $T$ and $T$ is a descendant of $S$, denoted by $an(T, \mathcal{G})$ and $de(S, \mathcal{G})$. If there is a possibly causal path from $S$ to $T$, then we say $S$ is a possible ancestor of $T$ and $T$ is a possible descendant of $S$, denoted by $possAn(T, \mathcal{G})$ and $possDe(S, \mathcal{G})$. As a convention, we regard every node as an ancestor and a descendant of itself.

**MPDAGs Construction.** Borrowed from [42], Algorithm 3 summarizes, the way to construct the maximal PDAG $\mathcal{G}'$ from the maximal PDAG $\mathcal{G}$ and backgroud knowledge $\mathcal{B}$, by leveraging Meek's rule in Figure 4. Specifically, here the background knowledge $\mathcal{B}$ is assumed to be the *direct causal information* in the form $S \to T$, meaning that $S$ is a direct cause of $T$. If Algorithm 3 does not return FAIL, then the background knowledge $\mathcal{B}$ and returned maximal PDAG $\mathcal{G}'$ are consistent with the input maximal PDAG $\mathcal{G}$.

---

**Algorithm 3** Construct MPDAG [34, 42]

---

1: **Inputs:** MPDAG $\mathcal{G}$ and Background knowledge $\mathcal{B}$.
2: **Output:** MPDAG $\mathcal{G}'$ or FAIL.
3: Let $\mathcal{G}' = \mathcal{G}$;
4: **while** $\mathcal{B} \ne \varnothing$ **do**
5:     Select an edge $\{S \to T\}$ in $\mathcal{B}$;
6:     $\mathcal{B} = \mathcal{B} \backslash \{S \to T\}$;
7:     **if** $\{S - T\}$ OR $\{S \to T\}$ is in $\mathcal{G}'$ **then**
8:         Orient $\{S \to T\}$ in $\mathcal{G}'$;
9:         Orienting edges in $\mathcal{G}'$ following the rules in Figure 4 until no edge can be oriented;
10:     **else**
11:         FAIL;

---

Figure 4: Meek's orientation rules: R1, R2, R3 and R4 [34]. For each rule, if the left-hand side graph is an induced subgraph of a PDAG $\mathcal{G}$, orient the undirected edge on it with the direction on the right-hand side.

## A.1 Existing results

**Lemma A.1.** *[42, Lemma B.1] Let $p = \langle V_1, ..., V_k \rangle$ be a b-possibly causal definite status path in an MPDAG $\mathcal{G}$. If there is a node $i \in \{1, ..., n - 1\}$ such that $V_i \to V_{i+1}$, then $p(V_i, V_k)$ is a causal path in $\mathcal{G}$.*

**Lemma A.2.** *[42, Lemma 3.6] Let $S$ and $T$ be distinct nodes in an MPDAG $\mathcal{G}$. If $p$ is a b-possibly causal path from $S$ to $T$ in $\mathcal{G}$, then a subsequence $p^*$ of $p$ forms a b-possibly causal unshielded path from $S$ to $T$ in $\mathcal{G}$.*

Let $X$ be a variable in an MPDAG $\mathcal{G}$, $\mathbf{R} \subset sib(X, \mathcal{G})$, then we use $\mathcal{G}_{\mathbf{R} \to X}$ to denote the partially directed graph resulted by orienting $\mathbf{R} \to X$ and $X \to sib(X, \mathcal{G}) \backslash \mathbf{R}$ in $\mathcal{G}$. Fang & He [15] propose the following Theorem A.3 to check the existence of $\mathcal{G}_{\mathbf{R} \to X}$.

**Theorem A.3.** *[15, Theorem 1] Let $\mathcal{G}$ be an MPDAG consistent with a CPDAG $\mathcal{G}^*$. For any vertex $X$ and $\mathbf{R} \subset sib(X, \mathcal{G})$, the following three statements are equivalent.*

- *There is a DAG $\mathcal{D} \in [\mathcal{G}]$ such that $pa(X, \mathcal{D}) = \mathbf{R} \cup pa(X, \mathcal{G})$ and $ch(X, \mathcal{D}) = sib(X, \mathcal{G}) \cup ch(X, \mathcal{G}) \backslash \mathbf{R}$.*

- *Compared with $\mathcal{G}$, $\mathcal{G}_{\mathbf{R} \to X}$ does not introduce any new V-structure collided on $X$ or any directed triangle containing $X$.*

- *The induced subgraph of $\mathcal{G}$ over $\mathbf{R}$ is complete, and there does not exist an $R \in \mathbf{R}$ and a $W \in adj(X, \mathcal{G}) \backslash (\mathbf{R} \cup pa(X, \mathcal{G}))$ such that $W \to R$.*

**Definition A.4** (Critical Set). *[15, Definition 2] Let $\mathcal{G}^*$ be a CPDAG. $S$ and $T$ are two distinct vertices in $\mathcal{G}^*$. The critical set of $S$ with respect to $T$ in $\mathcal{G}^*$ consists of all adjacent vertices of $S$ lying on at least one chordless possibly causal path from $S$ to $T$.*

**Theorem A.5.** *[16, Theorem 1] Suppose that $\mathcal{G}^*$ is a CPDAG, $S$ and $T$ are two distinct vertices in $\mathcal{G}^*$, and $\mathbf{C}$ is the critical set of $S$ with respect to $T$ in $\mathcal{G}^*$. Then, $T$ is a definite descendant of $S$ if and only if $\mathbf{C} \cap ch(S, \mathcal{G}^*) \neq \varnothing$, or $\mathbf{C}$ is non-empty and induces an incomplete subgraph of $\mathcal{G}^*$.*

**Lemma A.6.** *[32, Lemma 3.1] Given a CPDAG $\mathcal{G}^*$, a variable $X$, and $R \subset sib(X, \mathcal{G}^*)$, orienting $R \to X$ for each $R \in \mathbf{R}$ and $X \to W$ for each $W \in sib(S, \mathcal{G}^*) \backslash \mathbf{R}$ is consistent with $\mathcal{G}^*$ if and only if new orientations do not introduce v-structures collided on $X$.*

# B   Detailed proofs

## B.1   Proof of Lemma 4.3

*Proof.* First, we prove the sufficiency. Let $\mathcal{D}$ be any underlying DAG $\mathcal{D} \in [\mathcal{G}]$, and $\mathbf{C}$ be the critical set of $S$ with respect to $T$ in $\mathcal{G}$. Suppose $C \in \mathbf{C}$ is a child of $S$ in $\mathcal{D}$, that is $S \to C$ in $\mathcal{D}$. By the definition of critical set, $C$ lies on a chordless b-possibly causal path $\pi$ from $S$ to $T$ in $\mathcal{G}$. Since $S \to C$ in $\mathcal{D}$, by Lemma A.1, the corresponding path $\pi$ in $\mathcal{D}$ is directed. Therefore, $S$ is an ancestor of $T$ in the underlying DAG.

Next, we prove the necessity: For another direction, suppose that $S$ is a definite ancestor of $T$ in any underlying DAG $\mathcal{D}$. Let $\pi$ be the shortest causal path from $S$ to $T$ in $\mathcal{D}$, then the corresponding path of $\pi$ in $\mathcal{G}$ is a chordless b-possibly causal path, since if $\pi$ has any chord in $\mathcal{G}$, $\pi$ in $\mathcal{D}$ cannot be the shortest path. Denote the vertex adjacent to $S$ on $\pi$ be $C$, then $C \in \mathbf{C}$ and $C$ is a child of $S$ in the DAG $\mathcal{D}$. Therefore, if $T$ is definite descendant of $S$ in $\mathcal{G}$, then $\mathbf{C}$ always contains a child of $S$ in every DAG $\mathcal{D} \in [\mathcal{G}]$. □

## B.2   Proof of Lemma 4.4

The proof idea of Lemma 4.4 is to find the graphical condition in an MPDAG that when $\mathbf{C} \neq \varnothing$, all vertices in $\mathbf{C}$ or a superset of $\mathbf{C}$ can be oriented to $X$ in some Markov equivalent DAG, by utilizing locally valid orientation rules for MPDAGs (Theorem A.3). The rules are to check whether a set of variables in an MPDAG can be the parents of a given target.

In this section, we will first introduce some technical lemmas, and then prove Lemma 4.4 in Section 4.1.

### B.2.1   Technical lemmas

In this section, we introduce some technical lemmas that are useful in the proof of Lemma 4.4.

**Lemma B.1.** *Let $\mathcal{G}$ be an MPDAG. For any vertex $X$ in $\mathcal{G}$ and $\mathbf{R} \subset sib(X, \mathcal{G})$, if $\mathbf{R}$ induces a complete subgraph of $\mathcal{G}$ and there exists a $R \in \mathbf{R}$ and a $W \in adj(X, \mathcal{G}) \backslash (\mathbf{R} \cup pa(X, \mathcal{G}))$ such that $W \to R$, then $\mathbf{R} \cup W$ induces a complete subgraph of $\mathcal{G}$ and $W \in sib(X, \mathcal{G}) \backslash \mathbf{R}$.*

*Proof.* Suppose for a contradiction that $\mathbf{R} \cup W$ induces an incomplete subgraph of $\mathcal{G}$. Since $\mathbf{R}$ induces a complete subgraph of $\mathcal{G}$, that means some vertex $R' \in \mathbf{R}$ is not adjacent with $W$. As $W \in adj(X, \mathcal{G}) \backslash (\mathbf{R} \cup pa(X, \mathcal{G}))$, the node $W$ can be a child or a sibling of $X$.

(1) For the first case, as in Figure 5a, if $W$ is a child of $X$ in $\mathcal{G}$, since $W \rightarrow R$, then $X - R$ can be oriented by Rule 2 in Meek's criteria as $X \rightarrow R$, which contradicts that $R$ is a sibling of $X$;

(2) For the second case, if $W$ is a sibling of $X$ in $\mathcal{G}$ and $W \rightarrow R$, then the edge between $R$ and $R'$ can not be an undirected edge in $\mathcal{G}$, since $R'$ and $W$ are not adjacent. $R \rightarrow R'$ can be oriented by Meek's Rule 2, or $R \leftarrow R'$ and $\langle R', R, W \rangle$ is a v-structure collided on $R$. For the former case, if $R \rightarrow R'$ is in $\mathcal{G}$ as in Figure 5b, then $X \rightarrow R'$ can be oriented by Rule 4, which contradicts that $R'$ is a sibling of $X$ in $\mathcal{G}$. For the latter case in Figure 5c, if $W$ is a sibling of $X$ in $\mathcal{G}$ and $W \rightarrow R \leftarrow R'$, since $R'$ and $W$ are not adjacent, then $X - R$ can be oriented as $X \rightarrow R$ in $\mathcal{G}$ by Meek's Rule 3, which contradicts that $R$ is a sibling of $X$. Therefore, there does not exist any vertex $R' \in \mathbf{R}$ not adjacent with $W$, so $\mathbf{R} \cup W$ induces a complete subgraph of $\mathcal{G}$.

$\square$

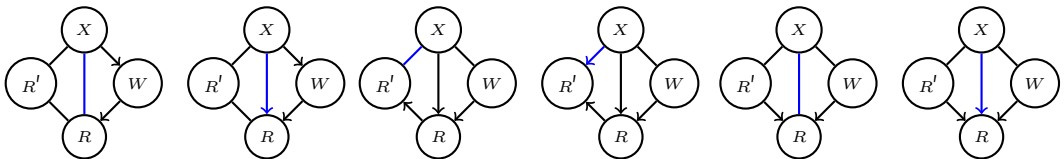

(a) $W$ is a child of $X$.    (b) $W$ is a sibling of $X$ and $R \rightarrow R'$.(c) $W$ is a sibling of $X$ and $R \leftarrow R'$.

Figure 5: The three cases discussed in the proof of Lemma B.1. For each case, the the blue undirected edge on the left-hand side subgraph will be oriented as the blue edge on the right-hand side subgraph.

**Lemma B.2.** *In an MPDAG $\mathcal{G}$, for any vertex $X$, there exists $\mathbf{H} \subseteq sib(X, \mathcal{G})$ that induces a complete subgraph of $\mathcal{G}$ if and only if there exists some $\mathbf{R}$ that $\mathbf{H} \subseteq \mathbf{R} \subseteq sib(X, \mathcal{G})$, such that there is a DAG $\mathcal{D} \in [\mathcal{G}]$ that $pa(X, \mathcal{D}) = \mathbf{R} \cup pa(X, \mathcal{G})$ and $ch(X, \mathcal{D}) = sib(X, \mathcal{G}) \cup ch(X, \mathcal{G}) \backslash \mathbf{R}$.*

*Proof.* According to Theorem A.3, in an MPDAG $\mathcal{G}$, for any vertex $X$ and $\mathbf{R} \subset sib(X, \mathcal{G})$, the following two statements are equivalent:

(1) There is a DAG $\mathcal{D} \in [\mathcal{G}]$ such that $pa(X, \mathcal{D}) = \mathbf{R} \cup pa(X, \mathcal{G})$ and $ch(X, \mathcal{D}) = sib(X, \mathcal{G}) \cup ch(X, \mathcal{G}) \backslash \mathbf{R}$.

(2) The induced subgraph of $\mathcal{G}$ over $\mathbf{R}$ is complete, and there does not exist an $R \in \mathbf{R}$ and a $W \in adj(X, \mathcal{G}) \backslash (\mathbf{R} \cup pa(X, \mathcal{G}))$ such that $W \rightarrow R$.

Therefore, Lemma B.2 can also be stated as: In an MPDAG $\mathcal{G}$, for any vertex $X$, there exists $\mathbf{H} \subseteq sib(X, \mathcal{G})$ that induces a complete subgraph of $\mathcal{G}$ if and only if there exists some $\mathbf{R}$ that $\mathbf{H} \subseteq \mathbf{R} \subseteq sib(X, \mathcal{G})$, such that the induced subgraph of $\mathcal{G}$ over $\mathbf{R}$ is complete, and there does not exist an $R \in \mathbf{R}$ and a $W \in adj(X, \mathcal{G}) \backslash (\mathbf{R} \cup pa(X, \mathcal{G}))$ such that $W \rightarrow R$.

The proof of sufficiency is straightforward. That set $\mathbf{R}$ induces a complete subgraph can ensure that any subset of $\mathbf{R}$ induces a complete subgraph of $\mathcal{G}$.

Next, we prove the necessity. If $\mathbf{H} \subseteq \mathbf{R}$ is complete and there does not exist an $H \in \mathbf{H}$ and a $W \in adj(X, \mathcal{G}) \backslash (\mathbf{H} \cup pa(X, \mathcal{G}))$ such that $W \rightarrow H$, then $\mathbf{H}$ satisfies the condition and we are done. Otherwise, if $\mathbf{H}$ is complete and there exists an $H \in \mathbf{H}$ and a $W \in adj(X, \mathcal{G}) \backslash (\mathbf{H} \cup pa(X, \mathcal{G}))$ such that $W \rightarrow H$, by Lemma B.1, $(\mathbf{H} \cup W) \subseteq \mathbf{R}$ induces a complete subgraph of $\mathcal{G}$. Similarly, if $\mathbf{H} \cup W$ is complete and there does not exist an $H' \in \mathbf{H} \cup W$ and a $W' \in adj(X, \mathcal{G}) \backslash (\mathbf{H} \cup W \cup pa(X, \mathcal{G}))$ such that $W' \rightarrow H'$, then $\mathbf{H} \cup W$ satisfies the condition and we are done. Otherwise, $(\mathbf{H} \cup W \cup W') \subseteq \mathbf{R}$ induces a complete subgraph of $\mathcal{G}$. Following this derivation, either we are done or we will end with the result that $sib(X, \mathcal{G})$ is complete. For the latter situation, by Theorem A.3, since orienting every vertex in $sib(X, \mathcal{G})$ towards $X$ does not introduce any new V-structure collided on $X$ or any directed triangle containing $X$. In this case, $\mathbf{R} = sib(X, \mathcal{G})$ meets the left hand side and we are done. $\square$

### B.2.2   Proof of Lemma 4.4

*Proof.* We first show the necessity. By Definition 4.2, $\mathbf{C} \subseteq sib(S, \mathcal{G}) \cup ch(S, \mathcal{G})$. Let $\mathcal{D} \in [\mathcal{G}]$ be an arbitrary DAG. If $\mathbf{C} \cap ch(S, \mathcal{D}) = \varnothing$ and $\mathbf{C} \neq \varnothing$, then $\mathbf{C} \subseteq pa(S, \mathcal{D})$, and thus $\mathbf{C} \subseteq sib(S, \mathcal{G})$. Denote by $\mathbf{R} = sib(S, \mathcal{G}) \cap pa(S, \mathcal{D})$, we have $\mathbf{C} \subseteq \mathbf{R} \subseteq sib(S, \mathcal{G})$ and $\mathbf{C} \subseteq \mathbf{R} \subseteq pa(S, \mathcal{D})$. Theorem A.3 proved that a non-empty subset $\mathbf{R}$ of $sib(S, G)$ can be a part of $S$'s parent set in some equivalent DAG if and only if $\mathbf{R}$ induces a complete subgraph of $\mathcal{G}$, and there does not exist a set $R \in \mathbf{R}$ and a $W \in adj(S, \mathcal{G}) \backslash (\mathbf{R} \cup pa(S, \mathcal{G}))$ such that $W \rightarrow R$. Therefore, as a subset of $\mathbf{R}$, $\mathbf{C}$ induces a complete subgraph of $\mathcal{G}$. This completes the proof of necessity.

We next prove the sufficiency. If $\mathbf{C} = \varnothing$, then it is straightforward that $\mathbf{C} \cap ch(S, \mathcal{D}) = \varnothing$ for some $\mathcal{D} \in [\mathcal{G}]$. Now assume $\mathbf{C} \neq \varnothing$ and $\mathbf{C} \cap ch(S, \mathcal{G}) = \varnothing$. As $\mathbf{C} \subseteq sib(S, \mathcal{G}) \cup ch(S, \mathcal{G})$, we have $\mathbf{C} \subseteq sib(S, \mathcal{G})$. Since $\mathbf{C}$ induces a complete subgraph of $\mathcal{G}$, by Lemma B.2, there exists $\mathbf{R}, \mathbf{C} \subseteq \mathbf{R} \subseteq sib(S, \mathcal{G})$, that there is a DAG $\mathcal{D} \in [\mathcal{G}]$ such that $pa(S, \mathcal{D}) = \mathbf{R} \cup pa(S, \mathcal{G})$ and $ch(S, \mathcal{D}) = sib(S, \mathcal{G}) \cup ch(S, \mathcal{G}) \backslash \mathbf{R}$. As $\mathbf{R} \subseteq pa(S, \mathcal{D})$ and $\mathbf{C} \subseteq \mathbf{R}$, $\mathbf{C} \subseteq pa(S, \mathcal{D})$ and thus $\mathbf{C} \cap ch(S, \mathcal{D}) = \varnothing$. $\qquad\square$

### B.3   Proof of Theorem 4.5

Theorem 4.5 is closely related to Theorem A.5 for CPDAGs from [16]. Since CPDAG is a special case of MPDAG, all results for MPDAGs works for CPDAGs. Although the condition provided by these two theorems are the same, they based on different theoretical results on locally valid orientation rules for CPDAGs and MPDAGs. The one for CPDAGs is mainly based on Lemma A.6 and for MPDAGs, it is mainly based on Theorem A.3.

*Proof.* Figure 6 shows how all lemmas fit together to prove the Theorem 4.5. To decide whether $T$ is a definite descendant of $S$ in an MPDAG $\mathcal{G}$, Lemma 4.3 provides a sufficient and necessary condition on a graphical characteristic of $\mathbf{C}$ on every DAG $\mathcal{D} \in [\mathcal{G}]$, which is then further explored by Lemma 4.4 to a graphical characteristic of $\mathbf{C}$ on the corresponding MPDAG $\mathcal{G}$. Following from Lemma 4.3 and Lemma 4.4, we have the desired sufficient and necessary condition to check whether $T$ is a definite descendant of $S$ in an MPDAG $\mathcal{G}$ on the graphical characteristic of $\mathbf{C}$. $\qquad\square$

$$\text{Lemma B.1} \longrightarrow \text{Lemma B.2} \longrightarrow \text{Lemma 4.4} \longrightarrow \textbf{Theorem 4.5} \longleftarrow \text{Lemma 4.3}$$

Figure 6: Proof structure of Theorem 4.5

### B.4   Proof of Lemma 4.6

*Proof.* Suppose $p = \langle S = V_0, ..., V_k = T \rangle$ is a chordless path from $S$ to $T$. We will show that $p$ is of definite status by showing that every vertex on $p$ is of definite status. Any triple $\langle V_{i-1}, V_i, V_{i+1} \rangle$ on $p$ with $i \in \{1, ..., k-1\}$ can be in the form: (1) $V_{i-1} \rightarrow V_i \leftarrow V_{i+1}$; or (2) $V_i \rightarrow V_{i+1}$ or $V_{i-1} \rightarrow V_i$ on $p$ or $V_{i-1} - V_i - V_{i+1}$ is a subpath of $p$ and $V_{i-1}$ is not adjacent with $V_{i+1}$. In the former case, $V_i$ is a collider; in the latter case, $V_i$ is a definite non-collider. The triple cannot be in the form $V_{i-1} \rightarrow V_i - V_{i+1}$ or $V_{i-1} - V_i \leftarrow V_{i+1}$, since the undirected edge can be oriented departs $V_i$ by Rule 1 in Meek's criterion. Therefore, every vertex on $p$ is of definite status. Thus, we completes the proof that $p$ is of definite status. $\qquad\square$

### B.5   Proof of Proposition 4.7

*Proof.* If all definite status b-possibly causal path from $S$ to $T$ are chordless, then by Lemma 4.6, $\mathbf{F_{ST}}$ is exactly $\mathbf{C_{ST}}$. Suppose that there are definite status b-possibly causal path from $S$ to $T$ with chords. We first prove that $\mathbf{C_{ST}} \subseteq \mathbf{F_{ST}}$. By the definition of critical set, for any $C \in \mathbf{C_{ST}}$, there is a chordless possibly causal path $p$ from $X$ to $Y$ on which $C$ is adjacent to $S$. By Lemma 4.6, $p$ is also a definite status b-possibly causal path from $S$ to $T$ without any chord. Therefore, as an adjacent vertex of $S$ on $p$, $C \in \mathbf{F_{ST}}$ as well. Since $C$ is an arbitrary vertex in $\mathbf{C_{ST}}$, $\mathbf{C_{ST}} \subseteq \mathbf{F_{ST}}$. Then we prove $\mathbf{F_{ST}} \subseteq \mathbf{C_{ST}}$. For any $F \in \mathbf{F_{ST}}$, there is a definite status b-possibly causal path $p^*$ from $S$ to $T$ in $\mathcal{G}$ that there is no chord with $S$ as an endpoint, on which $F$ is adjacent to $X$. By Lemma A.2, some subsequence of $p^*$ forms a chordless b-possibly causal path $p^{**}$ from $S$ to $T$. As on $p^*$, there is no chord with $S$ as an endpoint, the chordless b-possibly causal path $p^{**}$ must start with the edge

$S - F...$ or $S \rightarrow F...$. Therefore, $F \in \mathbf{C_{ST}}$. Since $F$ is an arbitrary vertex in $\mathbf{F_{ST}}$, $\mathbf{F_{ST}} \subseteq \mathbf{C_{ST}}$. This completes the proof of Proposition 4.7. $\qquad\square$

### B.6 Proof of Proposition 5.2

*Proof.* Suppose there is a possible descendant of $A$ in $\mathcal{G}$, which is denoted by $B$. Then there is a b-possibly causal path $p$ from $A$ to $B$. By Lemma A.2, a subsequence $p^*$ of $p$ forms a b-possibly causal unshielded path from $A$ to $B$. Suppose $p^* = \langle A = V_0, ..., V_k = B \rangle$, Assumption 5.1 implies $A \rightarrow V_1$. By Lemma A.1, $p^*$ is a causal path from $A$ to $B$ in $\mathcal{G}$. Therefore, $B$ is a definite descendant of $A$. $\qquad\square$

## C  An illustration example for Theorem 4.5

**Example.** Consider the MPDAG $\mathcal{G}$ in Figure 7 and the node $A$. We show the ancestral relations between $A$ and any other nodes. In $\mathcal{G}$, it is obvious that $B, C, D$ and $H$ are possible descendants of $A$. That is because $B$ is adjacent with $A$, and the critical set of $B$ with respect to $A$ is itself, which induces a complete subgraph of $\mathcal{G}$. The same conclusion can be drawn for $C, D$, and $H$. Node $E$ is also a possible descendant of $A$, since the chordless possibly causal path from $A$ to $E$ are $A - B - E$ and $A - C - E$, the critical set of $A$ with respect to $E$ is $\{B, C\}$. As the induced subgraph of $\mathcal{G}$ over $\{B, C\}$ is complete, by Theorem 4.5, $E$ is not a definite descendant of $A$, so it is a possible descendant of $A$. For $F$, the chordless possibly causal path from $A$ to $F$ are $A - B - F$, $A - C - F$ and $A - D - F$, thus the critical set of $A$ with respect to $F$ is $\{B, C, D\}$. Since the corresponding induced subgraph is incomplete, by Theorem 4.5, $F$ is a definite descendant of $A$.

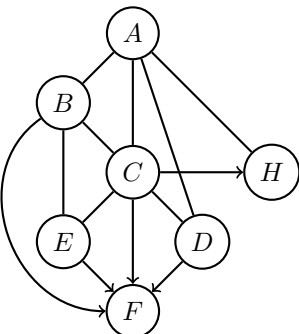

Figure 7: An MPDAG $\mathcal{G}$ for illustrating ancestral relations of the node $A$ with any other nodes. The node $B, C, D, E$ and $H$ are possible descendants of $A$; node $F$ is a definite descendant of $A$.

## D  Algorithm 1 and detailed explanation

In Algorithm 1, every b-possibly causal path of definite status (Line 12-13) on the way starting from $S$ to $\tau$ without any chord ending in $S$ (Line 14) is recorded in a queue $\mathcal{Q}$ as a triple $(\alpha, \phi, \tau)$, where $\alpha$ is the node lying immediately after $S$ and $\phi$ is the node that lie immediately before $\tau$ on the path. If $\tau$ is exactly $Y$, we add $\alpha$ to the critical set $\mathbf{C}$ and remove from $\mathcal{Q}$ all triples where the first element is $\alpha$, that is, we stop enumerating the required paths on which the node adjacent with $S$ is $\alpha$ (Line 8-9). Otherwise, we extend the path to the adjacencies of $\tau$, $\beta$, so that the path from $S$ to $\beta$ is still a b-possibly causal path of definite status without any chord ending in $S$ and then we add the corresponding triples to the queue $\mathcal{Q}$ (Line 11-16). In this algorithm, $\mathcal{H}$ is introduced to store the visited triples, and to avoid visiting the same triple twice.

## E  Algorithm in Section 5.1

**Algorithm 4** Identify the type of ancestral relation of $S$ with respect to all the other vertices in an MPDAG

1: **Input:** MPDAG $\mathcal{G}$, a variable $S$ in $\mathcal{G}$.
2: **Output:** The type of ancestral relation between $S$ and all the other vertices in $\mathcal{G}$.
3: **for** each node $W$ in $\mathcal{G}$ **do**
4:     Identify the type of ancestral relation of $S$ with respect to $W$ in $\mathcal{G}$ by Algorithm 2.
5: **Return** the set of definite descendants, possible descendants and definite non-descendants of $S$ in $\mathcal{G}$.

## F    Supplementary experimental results

### F.1    Causal graphs for one simulation

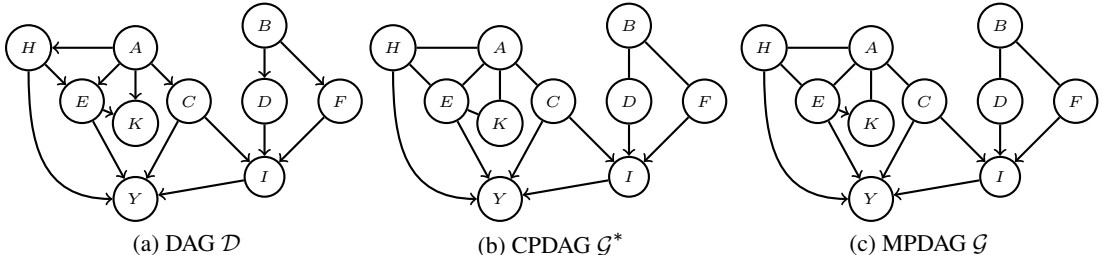

(a) DAG $\mathcal{D}$        (b) CPDAG $\mathcal{G}^*$        (c) MPDAG $\mathcal{G}$

Figure 8: (a) is one of the generated DAG $\mathcal{D}$ with 10 nodes and 10 directed edges; (b) is the corresponding CPDAG $\mathcal{G}^*$; (c) is the corresponding MPDAG $\mathcal{G}$ following Meek's rule, with the background knowledge that $E$ is a direct cause of $K$. The randomly selected sensitive attribute is represented by $A$ and the outcome attribute is $Y$. Algorithm 4 detects the ancestral relations in MPDAG $\mathcal{G}$: the definite non-descendants of the sensitive attributes are $\{B, D, F\}$, the possible descendants are $\{C, H, E, K, I\}$, and there is no definite descendants of $A$ in $\mathcal{G}$.

## F.2 Causal graphs for the Student Dataset

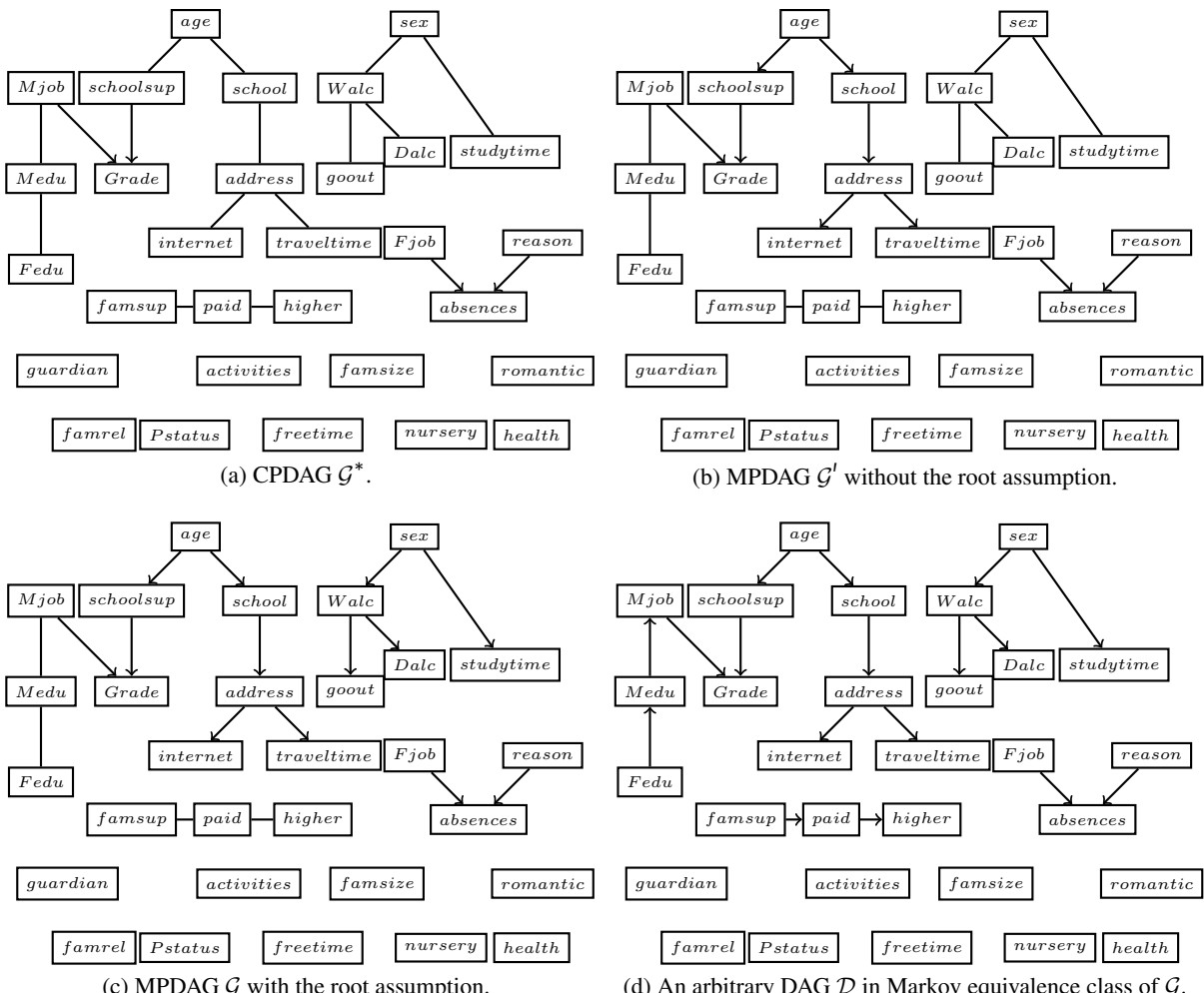

(a) CPDAG $\mathcal{G}^*$.

(b) MPDAG $\mathcal{G}'$ without the root assumption.

(c) MPDAG $\mathcal{G}$ with the root assumption.

(d) An arbitrary DAG $\mathcal{D}$ in Markov equivalence class of $\mathcal{G}$.

Figure 9: The causal graphs for Student dataset. The attribute information can be found at `https://archive.ics.uci.edu/ml/datasets/Student+Performance`. (a) is the learnt CPDAG $\mathcal{C}$; (b) Given the background knowledge that the age is the parent of *schoolsup* and *school*, without any other assumption, we can have the MPDAG $\mathcal{G}'$ by applying Meek's rule; (c) With the additional root node assumption, we can obtain the MPDAG $\mathcal{G}$; (d) is an arbitary DAG $\mathcal{D}$ in the Markov equivalent class of $\mathcal{G}$.

.

## F.3 Training details on real data

We assume the linear causal model in the obtained MPDAG $\mathcal{G}$. To test the counterfactual fairness of the baseline methods, as in Section 6.1, we first generate the counterfactual data. Since the ground-truth DAG is unknown, we generate the counterfactual data from a DAG sampled from the Markov equivalence class of MPDAG $\mathcal{G}$.[4] Then we fit the parameters of the model using the original data and generate samples from the model given the counterfactual *sex* and the same noise in the original data for each individual. We fit baseline models to both the original and counterfactual sampled data and measure the unfairness in the same way as in Section 6.1. This procedure is carried out 10 times and the average unfairness and RMSE results for five models are reported in Table 2.

---

[4]On this dataset, all the possible true DAGs give the same results as the nodes with uncertain edge directions are not related to the sensitive attribute.

## F.4 Experiment based on more complicated structural equations

To show the generality of our method, we generate each variable $X_i$ from the following non-linear structure equation:

$$X_i = g_i(f_i(pa(X_i) + \epsilon_i)), i = 1, ..., n, \qquad (2)$$

where the causal mechanism $f_i$ is randomly chosen from *linear*, *sin*, *cos*, *tanh*, *sigmoid* function and their combinations; $g_i$, which represents the post-nonlinear distortion in variable $X_i$, is randomly chosen from *linear* function, *absolute* and *reciprocal* function; $\epsilon_i$ is the noise term, sampling from *Gaussian*, *Exponential* and *Gumbel* distributions. With the same basic settings and evaluation metrics as Section 6.1, we fit a SVM regression model for the baselines and our proposed models. The average unfairness and RMSE achieved on 100 causal graphs is reported in Table 3. The corresponding boxplot is shown in Figure 10 as well. We can see that it yields the same trend on counterfactual fairness as the linear case, while the accuracy in this dataset does not necessarily decrease with the increase in fairness. More discussion on accuracy-fairness tradeoff can be referred to Appendix H.

Table 3: Average unfairness and RMSE for synthetic datasets generated by nonlinear structural equations on held-out test set. For each graph setting, the unfairness gets decreasing from left to right, while there is no obvious increase in RMSE.

|  | Node | Edge | Full | Unaware | FairRelax | Oracle | Fair |
|---|---|---|---|---|---|---|---|
| Unfairness | 10 | 20 | $0.575 \pm 0.431$ | $0.218 \pm 0.262$ | $0.028 \pm 0.115$ | $0.000 \pm 0.000$ | $0.000 \pm 0.000$ |
|  | 20 | 40 | $0.491 \pm 0.358$ | $0.143 \pm 0.210$ | $0.017 \pm 0.080$ | $0.000 \pm 0.000$ | $0.000 \pm 0.000$ |
|  | 30 | 60 | $0.388 \pm 0.309$ | $0.126 \pm 0.208$ | $0.010 \pm 0.044$ | $0.000 \pm 0.000$ | $0.000 \pm 0.000$ |
|  | 40 | 80 | $0.388 \pm 0.384$ | $0.094 \pm 0.139$ | $0.009 \pm 0.057$ | $0.000 \pm 0.000$ | $0.000 \pm 0.000$ |
| RMSE | 10 | 20 | $4.033 \pm 4.675$ | $4.024 \pm 4.663$ | $4.095 \pm 4.638$ | $4.098 \pm 4.649$ | $4.101 \pm 4.646$ |
|  | 20 | 40 | $3.921 \pm 4.532$ | $3.881 \pm 4.467$ | $3.921 \pm 4.497$ | $3.920 \pm 4.495$ | $3.927 \pm 4.491$ |
|  | 30 | 60 | $3.370 \pm 3.960$ | $3.371 \pm 3.958$ | $3.437 \pm 4.024$ | $3.438 \pm 4.025$ | $3.442 \pm 4.023$ |
|  | 40 | 80 | $3.457 \pm 3.999$ | $3.451 \pm 3.983$ | $3.474 \pm 3.956$ | $3.478 \pm 3.960$ | $3.479 \pm 3.963$ |

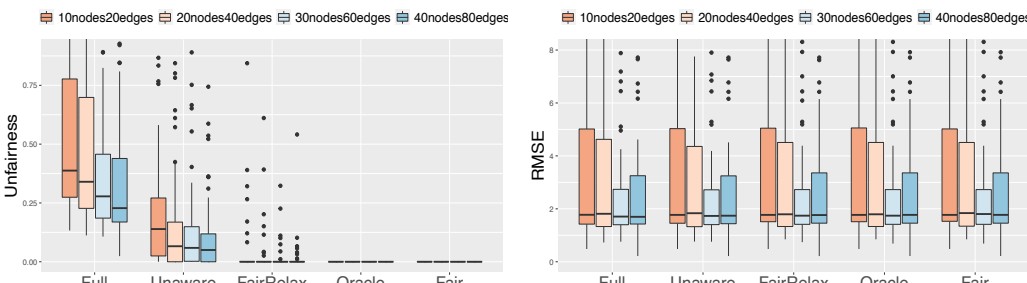

(a) Average unfairness for each model and graph setting. (b) Average RMSE for each model and graph setting.

Figure 10: Average unfairness and RMSE for synthetic datasets generated by nonlinear structural equations on held-out test set.

## F.5 Experiment analyzing fairness performance with varying amount of given domain knowledge

Prediction using `Fair` method results in a strictly fair model regardless of how much domain knowledge is given. Prediction using `FairRelax` method will show different fairness performance with different amount of domain knowledge, since some definite descendants $X$ of the sensitive attribute may be possible descendants when less domain knowledge is given, thus the unfair feature $X$ will be involved to make predictions. At this point we do not know, theoretically, how different amounts or even types of domain knowledge will affect the performance of `FairRelax`. However, we can explore this experimentally.

For a given CPDAG, only the fairness performance of `FairRelax` model among all models will be affected by the amount of background knowledge. The more background knowledge, the fairer the

`FairRelax`. For example, in the setting '10nodes20edges', when the proportion of the undirected edges' true orientation is increased from 0.1, 0.3, 0.6 to 1, the unfairness for `FairRelax` are 0.075, 0.023, 0.018, 0.0, respectively. The same trend can be found in other graph settings, see Table 4. The ' BK (%)' represents how much the undirected edges' true orientation is given as background knowledge.

Table 4: Average unfairness for synthetic datasets with varying amount of given domain knowledge. For each graph setting, the more domain knowledge, the fairer the model `FairRelax` becomes.

| Node | Edge | BK(%) | Full | Unaware | FairRelax | Oracle | Fair |
|---|---|---|---|---|---|---|---|
| | | 10 | $0.707 \pm 1.144$ | $0.587 \pm 1.093$ | $0.075 \pm 0.334$ | $0.0 \pm 0.0$ | $0.0 \pm 0.0$ |
| 10 | 20 | 30 | $0.707 \pm 1.144$ | $0.587 \pm 1.093$ | $0.023 \pm 0.178$ | $0.0 \pm 0.0$ | $0.0 \pm 0.0$ |
| | | 60 | $0.707 \pm 1.144$ | $0.587 \pm 1.093$ | $0.018 \pm 0.174$ | $0.0 \pm 0.0$ | $0.0 \pm 0.0$ |
| | | 100 | $0.707 \pm 1.144$ | $0.587 \pm 1.093$ | $0.000 \pm 0.174$ | $0.0 \pm 0.0$ | $0.0 \pm 0.0$ |
| | | 10 | $0.326 \pm 0.640$ | $0.280 \pm 0.624$ | $0.032 \pm 0.189$ | $0.0 \pm 0.0$ | $0.0 \pm 0.0$ |
| 20 | 40 | 30 | $0.326 \pm 0.640$ | $0.280 \pm 0.624$ | $0.018 \pm 0.136$ | $0.0 \pm 0.0$ | $0.0 \pm 0.0$ |
| | | 60 | $0.326 \pm 0.640$ | $0.280 \pm 0.624$ | $0.014 \pm 0.132$ | $0.0 \pm 0.0$ | $0.0 \pm 0.0$ |
| | | 100 | $0.326 \pm 0.640$ | $0.280 \pm 0.624$ | $0.000 \pm 0.000$ | $0.0 \pm 0.0$ | $0.0 \pm 0.0$ |
| | | 10 | $0.442 \pm 1.176$ | $0.433 \pm 1.191$ | $0.080 \pm 0.329$ | $0.0 \pm 0.0$ | $0.0 \pm 0.0$ |
| 30 | 60 | 30 | $0.442 \pm 1.176$ | $0.433 \pm 1.191$ | $0.076 \pm 0.321$ | $0.0 \pm 0.0$ | $0.0 \pm 0.0$ |
| | | 60 | $0.442 \pm 1.176$ | $0.433 \pm 1.191$ | $0.056 \pm 0.307$ | $0.0 \pm 0.0$ | $0.0 \pm 0.0$ |
| | | 100 | $0.442 \pm 1.176$ | $0.433 \pm 1.191$ | $0.000 \pm 0.000$ | $0.0 \pm 0.0$ | $0.0 \pm 0.0$ |
| | | 10 | $0.221 \pm 0.646$ | $0.199 \pm 0.647$ | $0.042 \pm 0.220$ | $0.0 \pm 0.0$ | $0.0 \pm 0.0$ |
| 40 | 80 | 30 | $0.221 \pm 0.646$ | $0.199 \pm 0.647$ | $0.019 \pm 0.176$ | $0.0 \pm 0.0$ | $0.0 \pm 0.0$ |
| | | 60 | $0.221 \pm 0.646$ | $0.199 \pm 0.647$ | $0.001 \pm 0.010$ | $0.0 \pm 0.0$ | $0.0 \pm 0.0$ |
| | | 100 | $0.221 \pm 0.646$ | $0.199 \pm 0.647$ | $0.000 \pm 0.000$ | $0.0 \pm 0.0$ | $0.0 \pm 0.0$ |

### F.6 Experiment analyzing model robustness on causal discovery algorithms

In Section 6.1, we obtain the ture CPDAG from the true DAG without running the causal discovery algorithm. However, in practice, with the true DAG unknown, the CPDAG can only be obtained from causal discovery algorithms. In order to test the model robustness on causal discovery algorithms, we learn the corresponding CPDAG from the synthetic data by the Greedy Equivalence Search (GES) procedure [5]. The prediction performance and fairness results are reported in Table 5 and Figure 11, from which, we can see the same trend on five models as the one in Section 6.1. Moreover, there is not much difference on fairness and prediction performance on `FairRelax` model between the case that the CPDAG is induced directly from the true DAG and the case that the CPDAG is learnt from the observational data by a causal discovery algorithm.

Table 5: Average unfairness and RMSE for synthetic datasets on held-out test set when the corresponding CPDAG is learned by GES search procedure. For each graph setting, the unfairness gets decreasing from left to right and the RMSE gets increasing from left to right.

| | Node | Edge | Full | Unaware | FairRelax | Oracle | Fair |
|---|---|---|---|---|---|---|---|
| **Unfairness** | 10 | 20 | $0.264 \pm 0.343$ | $0.203 \pm 0.318$ | $0.084 \pm 0.215$ | $0.000 \pm 0.000$ | $0.079 \pm 0.215$ |
| | 20 | 40 | $0.191 \pm 0.312$ | $0.150 \pm 0.283$ | $0.067 \pm 0.243$ | $0.000 \pm 0.000$ | $0.066 \pm 0.243$ |
| | 30 | 60 | $0.157 \pm 0.301$ | $0.143 \pm 0.308$ | $0.066 \pm 0.219$ | $0.000 \pm 0.000$ | $0.061 \pm 0.216$ |
| | 40 | 80 | $0.096 \pm 0.190$ | $0.074 \pm 0.183$ | $0.038 \pm 0.109$ | $0.000 \pm 0.000$ | $0.024 \pm 0.075$ |
| **RMSE** | 10 | 20 | $0.616 \pm 0.255$ | $0.631 \pm 0.262$ | $1.071 \pm 0.739$ | $1.079 \pm 0.788$ | $1.112 \pm 0.767$ |
| | 20 | 40 | $0.597 \pm 0.252$ | $0.601 \pm 0.250$ | $1.029 \pm 0.736$ | $0.862 \pm 0.564$ | $1.037 \pm 0.734$ |
| | 30 | 60 | $0.592 \pm 0.232$ | $0.595 \pm 0.235$ | $0.992 \pm 0.771$ | $0.907 \pm 0.894$ | $1.097 \pm 0.955$ |
| | 40 | 80 | $0.595 \pm 0.273$ | $0.596 \pm 0.272$ | $0.928 \pm 0.738$ | $0.746 \pm 0.433$ | $0.947 \pm 0.753$ |

## G  Additional related works on ancestral relations identifiability

A basic task in causal reasoning on an MPDAG $\mathcal{G}$ is to identify the ancestral relations between two distinct nodes in $\mathcal{G}$. The first intuitive method is to list all DAGs in $\mathcal{G}$ and then read off the ancestral

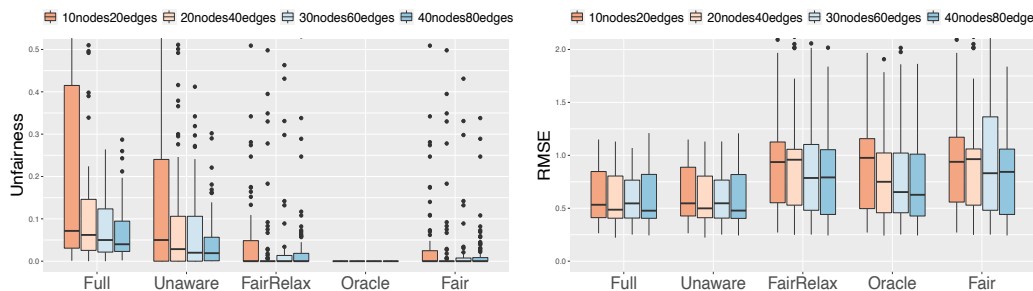

(a) Average unfairness for each model and graph setting. (b) Average RMSE for each model and graph setting.

Figure 11: Average unfairness and RMSE for synthetic datasets on held-out test set when the corresponding CPDAG is learned by GES search procedure. For each graph setting, the unfairness gets decreasing from left to right, while RMSE has the opposite trend.

relations in each DAG. However, this method is computationally burdensome. The second approach is to measure the possible causal effect [15, 21, 30, 31, 41, 42] from the source variable to the target variable in an MPDAG. The target is a definite descendant (or non-descendant) of the source if and only if all possible causal effect are non-zero (or zero). The third approach is to analyse the path from the source to target in an MPDAG $\mathcal{G}$. Perković et al. [42] propose that the target is a definite non-descendant of the source if and only if there is no b-possibly causal path from the source to target. There is also a sufficient and necessary graphical condition [16, Theorem 1] to identify whether a variable is a definite descendant of another variable in CPDAGs. The authors in [36, 47] [5] extend the sufficiency of this condition to other kinds of causal graphs as well. However, to the best of our knowledge, such graphical criterion to determine the definite descendants for MPDAGs has not been examined before.

## H   Discussion on accuracy-fairness trade-off

The accuracy-fairness trade-off is pointed out in a great number of existing algorithmic fairness works [2, 33, 35, 56, 70, 71]. Yet, accuracy may not be doomed to decrease as fairness increases depending on the data setting [13, 18, 61]. For example, in the synthetic dataset in Section 6.1, we do happen to observe an accuracy-fairness trade-off, while it seems that such trade-off does not exist in the synthetic dataset generated by the nonlinear structure equations in Appendix F.4. The authors in [13, 50, 57] describe when such trade-off exists and when it does not theoretically or empirically. Future work may take the fairness-accuracy trade-off into more consideration.

---

[5]Although Theorem 3.1 in [47] proved the necessity, their proof is incomplete as mentioned by [36]. Proving the necessity for more general types causal graphs remains an open problem [62].