# OpenReview forum: "Counterfactual Fairness with Partially Known Causal Graph"
_NeurIPS.cc/2022/Conference — NeurIPS 2022 Accept_

### Official Review · Reviewer_Hhvs · 2022-07-09

**Rating:** 6
**Confidence:** 4
**Soundness:** 3 good
**Presentation:** 3 good
**Contribution:** 3 good

**Summary:**

Given an MPDAG, the authors prove a method that can identify whether a variable is a definite descendant, definite non-descendant, or possible descendant of another variable. Building upon this result, the authors further study the problem of learning counterfactually fair models via selecting features.

**Questions:**

1. The introduction of b-critical set (Definition 4.2) is not necessary. Since a chordless b-possibly causal path from S to T definitely has no chord, it degenerates to a partially directed path. Therefore, there is no essential difference between the definition of b-critical set and the definition of critical set defined by Fang and He (2020), except that the latter is defined with respect to CPDAGs.

2. It seems to me that Lemma B.2 is flawed. Lemma B.2 states that if $R\subseteq sib(X, G)$ has a subset that induces a complete subgraph, then there is a DAG $D$ in $[G]$ such that $pa(X, D)=pa(X, G)\cup R$. If Lemma B.2 is correct, then for any $R\subseteq sib(X, G)$ that induces a complete subgraph, there is a DAG $D$ in $[G]$ such that $pa(X, D)=pa(X, G)\cup R$. However, this is impossible. For example, consider a complete graph with 4 variables: $X, S_1, S_2, C$. Every edge except $C\to S_1$ is undirected. Let $R=\\{S_1, S_2\\}$. $R$ induces a complete subgraph, but orienting $R\to X$ and $X \to C$ will lead to a directed cycle.

3. A minor point. Some cited papers have been published. Please revise the bib to make sure that their latest versions are cited.


**Limitations:**

The limitations of the paper have been addressed.

**Strengths And Weaknesses:**

Overall, I think this paper is relevant, new, and interesting. The paper is well-written and well-organized. The motivation is clearly described and the contribution is also clear. However, there is a technical lemma that seems flawed. Please refer to the detailed questions below.

---

> ### Author Response · Authors · 2022-08-02
> **Reply to Reviewer Hhvs**
>
> Thank you for your constructive comments. We address your comments point by point as follows. We have also revised the manuscript according to your suggestions.
>
> > The introduction of b-critical set (Definition 4.2) is not necessary. Since a chordless b-possibly causal path from $S$ to $T$ definitely has no chord, it degenerates to a partially directed path. Therefore, there is no essential difference between the definition of b-critical set and the definition of critical set defined by Fang and He (2020), except that the latter is defined with respect to CPDAGs.
>
> Thank you for reviewing so carefully and pointing this out. We agree that a chordless b-possibly causal path from $S$ to $T$ degenerates to a chordless possibly causal path and there is no essential difference between the definition of the b-critical set and the definition of the critical set defined by Fang and He (2020). Hence, we have removed the introduction of the b-critical set and replaced all 'b-critical set' by 'critical set' in the revision.
>
> > It seems to me that Lemma B.2 is flawed. Lemma B.2 states that if $\mathbf{R} \subseteq{sib(X,\mathcal{G})}$ has a subset that induces a complete subgraph, then there is a DAG $\mathcal{D} \in [\mathcal{G}]$ such that $pa(X,\mathcal{D})= pa(X,\mathcal{G}) \cup \mathbf{R}$. If Lemma B.2 is correct, then for any $\mathbf{R} \subseteq{sib(X,\mathcal{G})}$ that induces a complete subgraph, there is a DAG $\mathcal{D} \in [\mathcal{G}]$ such that $pa(X,\mathcal{D})= pa(X,\mathcal{G}) \cup \mathbf{R}$. However, this is impossible. For example, consider a complete graph with 4 variables: $X$, $S\_1$, $S\_2$, $C$. Every edge except $C \rightarrow S\_1$ is undirected. Let $R=\lbrace S_1,S_2 \rbrace$. $\mathbf{R}$ induces a complete subgraph, but orienting $\mathbf{R} \rightarrow X$ and $X \rightarrow C$ will lead to a directed cycle.
>
> Thanks for reading the supplementary material and being so careful. Yes, you are right. The previous statement for Lemma B.2 was easy to misunderstand, so we have rephrased Lemma B.2 in the revision. Lemma B.2 intended to say that if there is a set $\mathbf{H} \subseteq{sib(X,\mathcal{G})}$ inducing a complete subgraph, then there exists a superset $\mathbf{R}$ of $\mathbf{H}$ that $\mathbf{H} \subseteq{\mathbf{R}} \subseteq {sib(X,\mathcal{G})}$ such that there is a DAG $\mathcal{D} \in [\mathcal{G}]$ such that $pa(X,\mathcal{D})= pa(X,\mathcal{G}) \cup \mathbf{R}$. Therefore, in your example, let $\mathbf{H}= \lbrace S_1,S_2 \rbrace$ and $\mathbf{R}=\lbrace S_1,S_2,C \rbrace$. Orienting $\mathbf{H} \rightarrow X$ and $X \rightarrow C$ will lead to a directed cycle, but there still exists a superset $\mathbf{R}$ of $\mathbf{H}$ that $\mathbf{H} \subseteq \mathbf{R} \subseteq {sib(X,\mathcal{G})}$ such that orienting $\mathbf{R} \rightarrow X$ and $X \rightarrow \emptyset$ is reasonable, since it will not lead to a directed cycle or a collider, which also means there is a DAG $\mathcal{D} \in [\mathcal{G}]$ such that $pa(X,\mathcal{D})= pa(X,\mathcal{G}) \cup \mathbf{R}$ according to Theorem 1 in Fang and He (2020).
>
> The rephrased Lemma B.2 is as follows:
>
> - In an MPDAG $\mathcal{G}$, for any vertex $X$, there exists $\mathbf{H} \subseteq{sib(X,\mathcal{G})}$ that induces a complete subgraph of $\mathcal{G}$ if and only if there exists some $\mathbf{R}$ that $\mathbf{H} \subseteq \mathbf{R} \subseteq{sib(X,\mathcal{G})}$, such that there is a DAG $\mathcal{D} \in [\mathcal{G}]$ that $pa(X,\mathcal{D})=\mathbf{R} \cup pa(X,\mathcal{G})$ and $ch(X,\mathcal{D})=sib(X,\mathcal{G}) \cup ch(X, \mathcal{G}) \backslash \mathbf{R}$.
>
> Hope this addresses your concern and please kindly let us know if you have further concern.
> > A minor point. Some cited papers have been published. Please revise the bib to make sure that their latest versions are cited.
>
> Thanks for pointing this out. We have fixed this issue in the revision.

---

> > ### Comment · Reviewer_Hhvs · 2022-08-06
> > **Reply to the Authors**
> >
> > Thank you for your reply. All my concerns have been addressed and the revised Lemma B.2 is correct to me now.

---

> > > ### Author Response · Authors · 2022-08-08
> > > **Reply to Reviewer Hhvs**
> > >
> > > We feel fortunate that you are interested in the technical details of our work. Thanks again for strengthening our paper, and we greatly appreciate your efforts!
> > >
> > > Authors of 1219

---

> ### Author Response · Authors · 2022-08-05
> **Reply to Reviewer Hhvs**
>
> Dear Reviewer Hhvs,
>
> We appreciate your comments and time! We have provided answers to your questions and revised the paper following your suggestions. Would you mind checking it and confirming if you have further questions?
>
> Best Regards,
>
> Authors of 1219

---

### Official Review · Reviewer_UUSP · 2022-07-13

**Rating:** 3
**Confidence:** 4
**Ethics Flag:** Yes
**Soundness:** 2 fair
**Presentation:** 2 fair
**Contribution:** 2 fair

**Summary:**

This paper works within a structural causal modeling (SCM) framework for defining fairness of algorithms. Previous literature defining fairness this way assumes knowledge of the relevant SCM, which is a key limitation. The current paper relaxes this assumption to some specific contexts where the SCM is partially known, that is when a maximally partially directed acyclic graph (MPDAG) is available rather than a full directed acyclic graph (DAG). In the MPDAG framework a variable can be a definite descendent, definite non-descendent, or a possible descendent of other variables. This paper provides algorithms for determining which of these relationships holds for a given pair of variables and theorems to establish the correctness of the algorithms. The application to fairness involves considering which variables are definite or possible descendants of a sensitive attribute, and proposing corresponding definitions of fairness based on whether only definite non-descendants are used in the prediction (Fair) or possible descendants are also included (FairRelax) and only definite descendants are excluded.


**Questions:**

Q1: Counterfactual fairness is possible even with definite descendants provided counterfactual changes “cancel out” in the prediction. The current paper gives an incorrect definition on lines 51-53 which is based on a sufficient (but not necessary) condition for counterfactual fairness.

Q2: Should the unfairness of Oracle and Fair methods in the simulations be exactly zero, even in samples? I would think it will be close to zero in each realization and very close to zero on average, with small standard errors- but not necessarily exactly zero.

Q3: Most of the methods proposed in this paper do not seem to be necessarily connected to fairness. Presumably it is of more general interest to determine the ancestral relationships in an MPDAG. Would it make more sense to reframe the paper as a general method within the SCM literature, and its application to fairness just one of the motivating examples? This may also strengthen the paper since, as I point out in Q1, the connection of the current methods to fairness is based on a sufficient condition which may not be necessary, and hence it may not be a strong enough connection for fairness to be a main focus of the paper.

Minor issue: In section 5.1, isn’t the inclusion of “(possibly small)” a bit of wishful thinking? The violation of fairness could also possibly be large. Without knowledge of a specific context to justify either possibility perhaps this speculation should be omitted.

Minor issue: citing preprints instead of published versions, e.g. of [21, 23, 56]


**Ethics Review Area:**

["Discrimination / Bias / Fairness Concerns"]

**Limitations:**

The weaknesses and questions above, as well as the limitations inherited by any SCM type of approach which include untestable assumptions. The current paper also acknowledges its assumptions of no confounding or selection bias, but it is reasonable to try to address that in future work rather than all at once.

**Strengths And Weaknesses:**

Strength: SCM approaches rely on very strong assumptions so it is useful to relax these.

Strength: The simulation results use fairly high dimensional DAGs compared to the common examples considered in counterfactual fairness.

Weakness: Many of the results and basically all of the illustrative figures are in appendices. I think the paper could be improved by reorganization that includes showing examples in the main text, like figures 2, 3, and 6 which are currently in the appendices. As it is now, a reader who is not already familiar with PDAG, CPDAG, MPDAG will not know how to visualize these unless they read the supplementary material, and will struggle to understand the definitions and lemmas.

Weakness: It is possible the technical contribution of the current paper is a relatively small increment, extending Lemma 4.4 from previous work on CPDAGs. Since much of the details is in the appendix I did not review it all carefully enough to compare with the previous papers. This paper could be strengthened by clarifying which parts are new contributions.

---

> ### Author Response · Authors · 2022-08-02
> **Reply to Reviewer UUSP (2/2)**
>
> > Q2: Should the unfairness of Oracle and Fair methods in the simulations be exactly zero, even in samples? I would think it will be close to zero in each realization and very close to zero on average, with small standard errors- but not necessarily exactly zero.
>
> Yes, the unfairness of "Oracle" and "Fair" methods in the simulations is exactly zero for each realisation. This is because the "Oracle" and "Fair" models make predictions with non-descendants given the ground-truth DAG and definite non-descendants of the sensitive attribute in an MPDAG, respectively. These attributes have the exact same value in the counterfactual data as they do in the observational data, as they are unaffected by the sensitive attribute. Therefore, both the "Oracle" and "Fair" models will yield the same prediction for each individual over the observational and counterfactual data. We have provided the source code in the supplementary, and we would appreciate it if you could try out the code to verify the correctness of the simulations.
>
> > Q3: Most of the methods proposed in this paper do not seem to be necessarily connected to fairness. Presumably, it is of more general interest to determine the ancestral relationships in an MPDAG. Would it make more sense to reframe the paper as a general method within the SCM literature, and its application to fairness just one of the motivating examples? This may also strengthen the paper since, as I point out in Q1, the connection of the current methods to fairness is based on a sufficient condition which may not be necessary, and hence it may not be a strong enough connection for fairness to be a main focus of the paper.
>
> Thanks for your kind suggestion.
>
> - We agree that the identifiability of ancestral relations in an MPADAG could have other potential applications. We had the same idea as you when we finished the first draft, but we finally gave up this idea because we could not think of other applications with real impacts.
>
> - Here we would like to briefly summarise the motivation of our work (Line 37-63 in the first manuscript) to clarify our paper organisation further. The seminal work [Kusner et al., 2017] our work built on assumes that the DAG is known and thus one can easily determine the ancestral relations in a DAG to achieve counterfactual fairness. However, when the DAG is unknown, the causal discovery algorithms with background knowledge can only give us MPDAGs in most situations. To achieve counterfactual fairness in this situation, one naturally needs algorithms to determine the ancestral relations in MPDAGs. This is why we put much effort into determining the ancestral relations in MPDAGs. Moreover, as we have illustrated in section 5.2, the background knowledge of sensitive attributes as root nodes is very specific to the counterfactual fairness problem.
>
> - Taking into account your suggestions, we have added a short discussion in the conclusion part to remind the readers that our ancestral relation determining procedure in MPDAGs may have more applications. As for your concern on the sufficient condition of the counterfactual fairness, see our response to Q1. Hope this addresses your concern and please kindly let us know if you have further concerns.
>
> > Minor issue: In section 5.1, isn’t the inclusion of “(possibly small)” a bit of wishful thinking? The violation of fairness could also possibly be large. Without knowledge of a specific context to justify either possibility perhaps this speculation should be omitted.
>
> We have removed "(possibly small)" to make the statement more rigorous. As you have nicely noticed in Q1, if the parameters of descendants happen to cancel out, the violation would be small. Moreover, as we did not use the definite descendants, the chance of violation would be even smaller. This is also related to the accuracy-fairness trade-off as we have referenced in the Experiment section (Line 321) and discussed in Appendix H in the original manuscript.
>
> > Minor issue: citing preprints instead of published versions, e.g. of [21, 23, 56]
>
> Thanks for pointing this out. We have fixed this issue in the revision.

---

> > ### Comment · Reviewer_UUSP · 2022-08-09
> > **Still have significant concerns with the paper**
> >
> > I appreciate the authors' careful replies and I am glad they found some of my comments useful. However, I am sorry to say this exchange did not influence me to increase my initial rating, and I still have significant concerns with the paper.
> >
> > First, my previous points about the weak connection with fairness are unaddressed. The paper would be significantly stronger if it were not framed as specific to counterfactual fairness. There are two reasons for this, both of which I mentioned before:
> >
> > (1) The focus on using non-descendants is based entirely on a sufficient condition which is not necessary, and--I will emphasize--is extremely unlikely to hold in any real world applications where fairness is important. This is because sensitive attributes are likely causally influential on every measured variable of interest. Histories of economic, social, and cultural disadvantages/oppression mean we are unlikely to find any predictors of some important outcome which have not also been influenced by that unfair history. I think this is a realistic version and it's what I will stick with, but to make the point more clear in a more critical version: the contribution of this paper to fairness is an algorithm for finding variables satisfying some condition which essentially no variables will satisfy in real applications.
> >
> > (2) The ancestral closure assumption used in counterfactual fairness is a structural assumption that could reasonably extend to other applications. Hence, the contribution of the current paper could be applicable to a much wider scope. The methodological results here should be presented in the appropriate generality and abstraction, and doing so would help address the issue (1) above because then its applicability would not rely entirely on one setting where its usefulness may be too unrealistic.
> >
> > Second, even if we suppose the previous issue has been addressed, I retain my initial assessment that the paper provides relatively incremental technical progress on previous work. I do think the method could be useful and have some valuable applications, but I do not think it meets the standards of originality and impact expected of papers in such a highly selective conference as NeurIPS. (I hope the authors will not find this assessment discouraging because it is important to do valuable work like this whether or not it gets accepted at any one particular venue)

---

> > > ### Author Response · Authors · 2022-08-09
> > > **Further clarification to Reviewer UUSP**
> > >
> > > Thank you for your feedback.
> > >
> > > First, without any offense, we would like to point out that your statement "sensitive attributes are likely causally influential on every measured variable of interest" is purely subjective without any scientific evidence. If you insist your point is correct, we would appreciate it if you could provide us a real example where sensitive attributes causally influence every variable. Here we can provide you a counterexample. In the Student Performance Data Set used in our paper, only 4 out of 30 features are descendants of the sensitive attribute ‘sex’ according to the causal discovery algorithms. While the causal discovery algorithms could make mistakes, the results are surely stronger than your subjective judgement.
> > >
> > > Second, even if your subjective judgement is correct, the limitation of finding non-descendants originates from the seminal counterfactual fairness paper [Kusner et al., 2017]. I do not think our work should be blamed for not fixing this issue. Our work makes counterfactual fairness more practical by dropping the requirement of known causal graphs. This is a significant advancement, which has been extensively discussed in our paper and has been appreciated by the other two reviewers. This is also how research has progressed, with each work extending existing works toward certain direction, and ultimately the technology can be applied to a wider range of fields.
> > >
> > > Third, thanks for discovering that our technical contribution of identifiability of causal relations in MPDAG has potentially wider applications. However, again, we do not think we should be blamed for only applying it to counterfactual fairness. The identifiability of causal relations in MPDAG is strongly motivated by achieving more practical counterfactual fairness. In fact, many mathematical discoveries and techniques are designed for specific problems and the abstracted idea is then used for other problems. For example, the Monte Carlo methods were originally developed for particle physics and nowadays people in statistics and machine learning use this technique extensively.
> > >
> > > Last but not least, even if you subjectively dislike our work, we cannot understand why you give us a rating 3, which should be used for "a paper with technical flaws, weak evaluation, inadequate reproducibility and incompletely addressed ethical considerations". We understand and appreciate your effort to keep NeurIPS high standard, but we would appreciate more if you can base your assessment on solid evidence.
> > >
> > > Many thanks,
> > >
> > > Authors of paper 1219

---

> > > > ### Comment · Reviewer_UUSP · 2022-08-09
> > > > **What it would take for me to raise my rating**
> > > >
> > > > First I will reply regarding the rating, then move to the other important points. To quote from the reviewer guidelines:
> > > >
> > > > > 4: Borderline reject: Technically solid paper where reasons to reject, e.g., limited evaluation, outweigh reasons to accept, e.g., good evaluation. Please use sparingly.
> > > >
> > > > >  3: Reject: For instance, a paper with technical flaws, weak evaluation, inadequate reproducibility and incompletely addressed ethical considerations.
> > > >
> > > > Note that the description for 3 begins with "For instance," hence it does not mean the listed conditions are exhaustive. Also note that the borderline reject/accept descriptions ask us to "Please use sparingly." I would consider increasing my rating to one of these borderline ratings, despite my second point in the previous comment (about incremental technical progress), if the authors had addressed my other questions. But I am not satisfied with the responses to my other questions, as should be clear from this continued discussion, which I turn to next.
> > > >
> > > > About sensitive attributes being causally related to everything:
> > > >
> > > > > "sensitive attributes are likely causally influential on every measured variable of interest" is purely subjective without any scientific evidence
> > > >
> > > > On the contrary, this is one of the most persistent findings in social and health sciences. Providing one reference for it would almost be absurd because it requires something more like citing several entire disciplines. Taking race as an example, there are many references here https://en.wikipedia.org/wiki/Racial_inequality_in_the_United_States regarding the effects of racism on housing, education, health care, employment, wealth, and policing, and each of these things in turn (housing, education, health care, employment, etc) is also causally important for almost every other outcome we would study (and to each other).
> > > >
> > > > About the focus on using non-descendants:
> > > >
> > > > I do not believe the [Kusner et al., 2017] paper intended to establish its Lemma 1 as a standard or default approach. For one thing, they named this result a lemma, and for another see their section 4.2:
> > > >
> > > > > Level 1. Build $\hat Y$ using only the observable non-descendants of A. This only requires partial causal ordering and no further causal assumptions, **but in many problems there will be few, *if any*, observables which are not descendants of protected demographic factors.**
> > > >
> > > > I added the emphasis in this quote. They go on to give 2 other levels of assumptions which enable the construction of predictors that can depend on descendants (provided the dependence cancels out).
> > > >
> > > > Concluding:
> > > >
> > > > I see this work in its current stage as a classic "solution in search of a problem," and the application to counterfactual fairness (and specifically the version using an unrealistic sufficient condition) is not a good enough problem for me to believe the contribution is significant enough for this conference. As I mentioned earlier, if this issue had been addressed I would consider raising my rating.

---

> > > > > ### Author Response · Authors · 2022-08-09
> > > > > **Further Reply to Reviewer UUSP**
> > > > >
> > > > > Thank you for your quick reply.
> > > > >
> > > > > 1. First of all, without any offense, we would like to say that the Wikipedia link is not convincing, not to mention that it is only about race and lists only limited number of variables. Furthermore, how would you interpret the real data experiment in our paper?
> > > > >
> > > > > 2.
> > > > >
> > > > > > “I do not believe the [Kusner et al., 2017] paper intended to establish its Lemma 1 as a standard or default approach.” “They go on to give 2 other levels of assumptions which enable the construction of predictors that can depend on descendants (provided the dependence cancels out).”
> > > > >
> > > > > We respectively disagree with this point. The fact is that Lemma 1 in [Kusner et al., 2017] is indeed the standard of their methods. All three levels of assumptions in [Kusner et al., 2017] are based on its Lemma 1. Two other levels of assumptions cannot depend on descendants as well, which is discussed in section 4.2 and illustrated explicitly in Figure 2. The latent variable in level 2 and the error term in level 3 are both non-descendants of the sensitive attributes.
> > > > >
> > > > > Besides, it is obvious that the authors do not encourage to use descendants in prediction, as the explanation of Lemma 1 in section 3.2,
> > > > > > This does not exclude using a descendant W of A as a possible input to $\hat{Y}$. However,  **this will only be possible** in the case where the overall dependence of $\hat{Y}$ on A disappears, **which will not happen in general.** Hence, Lemma 1 provides the most straightforward way to achieve counterfactual fairness.
> > > > >
> > > > > 3.
> > > > >
> > > > > > Level 1. Build $\hat{Y}$ using only the observable non-descendants of A. This only requires partial causal ordering and no further causal assumptions, but **in many problems** there will be few, if any, observables which are not descendants of protected demographic factors.
> > > > >
> > > > > First, note that it is **"in many problems"** here, not "most". There is still many problems that can have quite a few non-descendants, and our real data is an example. Second, we would like to emphasize that the author's intention in section 4.2 is to use latent variable when there are few observables, but never mention or encourage to use descendants in prediction.

---

> > > > > > ### Comment · Reviewer_UUSP · 2022-08-10
> > > > > > **This is not a debate**
> > > > > >
> > > > > > I don't wish to quote and respond line by line trying to challenge every detail, that is counter productive and loses sight of the big picture. As a reviewer I am providing my judgment about the work, so I begin by reiterating my central message:
> > > > > >
> > > > > > > **I see this work in its current stage as a classic "solution in search of a problem," and the application to counterfactual fairness (and *specifically the version using an unrealistic sufficient condition*) is not a good enough problem** for me to believe the contribution is significant enough for this conference.
> > > > > >
> > > > > > I also want to reiterate the positive parts of my judgment that I think this contribution could be valuable for some *other* applications. However, the paper is framed entirely around counterfactual fairness, such that *even of the title of the paper does not indicate to potential readers that it contains a more general purpose causal discovery algorithm!* I would increase my rating if the authors changed this framing, but they have argued against that.
> > > > > >
> > > > > > It is not my responsibility to convince the authors that decades of social science research cannot simply be dismissed as follows:
> > > > > >
> > > > > > > the Wikipedia link is not convincing
> > > > > >
> > > > > > The link contains references to dozens of papers for each of the topics of education, housing, health, wealth, etc. The professional necessity of publishing an algorithm in a computer science conference does not justify simply dismissing other entire fields of study.
> > > > > >
> > > > > > I linked to that wikipedia article only as an example, but I would predict that if we choose almost any machine learning task where people care about "fairness" with respect to any "sensitive attribute," and if we go do a literature review on the social science related to that sensitive attribute, we will find many empirical studies establishing relationships between that attribute and causal pathways to the other observed variables we want to use. As the [Kusner et al., 2017] paper points out, we should care about such "domain knowledge":
> > > > > >
> > > > > > > Level 2. Postulate background latent variables that act as non-deterministic causes of observable variables, based on explicit domain knowledge...
> > > > > >
> > > > > > And as some of the same authors wrote elsewhere https://www.nature.com/articles/d41586-020-00274-3
> > > > > >
> > > > > > > Researchers in statistics and machine learning need to know more about the causes of unfairness in society. They should work closely with those in disciplines such as law, social sciences and the humanities.
> > > > > >
> > > > > > The same authors have written here https://arxiv.org/abs/1805.05859 that
> > > > > >
> > > > > > > Despite its desirable features, **there are major prices to be paid** by avoiding structural equations. For technical reasons, enforcing such constraints require throwing away information from any descendant of A that is judged to be on an “unfair path” from A to Y .
> > > > > >
> > > > > > My point is that those authors are clearly not arguing for staying at what they call "Level 1" as some kind of default, standard approach. They have written multiple times about the importance of doing more explicit modeling that incorporates domain knowledge, i.e. levels 2+. So let's return to those levels and your claim regarding them:
> > > > > >
> > > > > > > Two other levels of assumptions cannot depend on descendants as well, which is discussed in section 4.2 and illustrated explicitly in Figure 2. The latent variable in level 2 and the error term in level 3 are both non-descendants of the sensitive attributes.
> > > > > >
> > > > > > This is a misunderstanding. It is not generally possible to do Levels 2 and 3 without using descendants of A, because these descendants may need to be used to estimate the latent factors. For example, see where they use Level 3 in their section 5:
> > > > > >
> > > > > > > We estimate the error terms by first fitting two models that each **use race and sex** to individually predict GPA and LSAT. We then compute the residuals ... We use these residual estimates ... to predict FYA. We call this Fair Add
> > > > > >
> > > > > > So their Level 3 fair predictor is a function that explicitly uses the sensitive attributes and their descendants.
> > > > > >
> > > > > > Finally, I initially did not respond to the point below because I wanted to keep my discussion focused on the key reasons for my judgment, but since it is now being reiterated I will respond.
> > > > > >
> > > > > > > Furthermore, how would you interpret the real data experiment in our paper?
> > > > > >
> > > > > > It is irrelevant because (1) we do not know the "ground truth" in this data to compare the results to, and (2) standard assumptions like no selection bias and no confounding will almost never hold on any real dataset.

---

> ### Author Response · Authors · 2022-08-02
> **Reply to Reviewer UUSP (1/2)**
>
> Thank you for your helpful comments. We address your comments point by point as follows.  We have also tried our best to revise the manuscript according to your suggestions.
>
> > Weakness: Many of the results and basically all of the illustrative figures are in appendices. I think the paper could be improved by reorganization that includes showing examples in the main text, like figures 2, 3, and 6 which are currently in the appendices. As it is now, a reader who is not already familiar with PDAG, CPDAG, MPDAG will not know how to visualize these unless they read the supplementary material, and will struggle to understand the definitions and lemmas.
>
> Thanks for your nice suggestion. Given that NeurIPS is a technical conference with a page limit for the submissions, we have to assume that the readers already have some background in causality so that we can focus on presenting our new results. In case some readers do not have a technical background, we have tried to include more preliminaries in Appendix A.  We will move some of the suggested figures to the main paper in the camera ready version as one additional page will be allowed. The reason we put Figure 3 in the appendix is that it is used in the proof procedure of Lemma B.1.
>
> > Weakness: It is possible the technical contribution of the current paper is a relatively small increment, extending Lemma 4.4 from previous work on CPDAGs. Since much of the details is in the appendix I did not review it all carefully enough to compare with the previous papers. This paper could be strengthened by clarifying which parts are new contributions. Hope this addresses your concern and please kindly let us know if you have further concerns.
>
> Again, sorry for not being able to put the proof details in the main text. Below we provide a summary of technical novelties compared to the work on CPDAG. CPDAG is a special case of MPDAG (Line 110 and 676 in the original manuscript), and MPDAG can represent more information regarding the causal relationships than CPDAGs. However, this additional information has not been fully exploited in practice due to the fact that causal methods that are applicable to CPDAGs are not directly applicable to general MPDAGs. For example, the CPDAGs can have partially directed cycles while MPDAGs cannot. Lemma 4.4 is therefore not a straightforward extension from CPDAG. To establish Lemma 4.4, we introduced two new technical lemmas (Lemma B.1 and Lemma B.2) exploring the properties of the general MPDAGs.
>
> As the novelty lies in the proof techniques, we have attempted to explain the key difference and difficulty in the proof of the desired Theorem 4.5 and the one for CPDAGs on Line 676-680 of the appendix in the original manuscript. To better explain our technical contributions in identifying ancestral relations in MPDAGs in the main, we have added a discussion on difficulties and differences in working with MPDAGs, as opposed to CPDAGs in the revision. Hope this addresses your concern and please kindly let us know if you have further concerns.
>
>  > Q1: Counterfactual fairness is possible even with definite descendants provided counterfactual changes “cancel out” in the prediction. The current paper gives an incorrect definition on lines 51-53 which is based on a sufficient (but not necessary) condition for counterfactual fairness.
>
> Nice thinking! Yes, it is possible with definite descendants provided counterfactual changes “cancel out” in the prediction. Our statement of counterfactual fairness on Line 51-53 exactly follows the pioneering counterfactual fairness work [Kusner et al., 2017, Lemma 1], which is indeed based on a sufficient (but not necessary) condition for counterfactual fairness. However, we would like to argue that the sufficient condition is reasonable, because the "cancel out" situation can only happen for a special set of model parameters. Lemma 1 in [Kusner et al., 2017] is more strict in the sense that fairness is a property of the graph and works for all possible parameters. Extension of this implication of the counterfactual fairness notion is definitely an interesting direction, which is beyond the scope of our paper.

---

> ### Author Response · Authors · 2022-08-05
> **Reply to Reviewer UUSP**
>
> Dear Reviewer UUSP,
>
> We appreciate your comments and time! We have provided answers to your questions and revised the paper following your suggestions. Would you mind checking it and confirming if you have further questions?
>
> Best Regards,
>
> Authors of 1219

---

> ### Author Response · Authors · 2022-08-07
> **Reply to Reviewer UUSP**
>
> Dear Reviewer UUSP,
>
> We appreciate your comments and time! Half of the author-reviewer discussion time has passed, and we wonder whether you need further clarification. We look forward to hearing from you, and we are happy to discuss anything with you about our work!
>
> Best regards,
>
> Authors of 1219

---

> ### Author Response · Authors · 2022-08-08
> **Reply to Reviewer UUSP**
>
> Dear Reviewer UUSP,
>
> Thanks for reviewing our paper. This is the last day for the Author-Reviewer Discussion period. We understand that your are very busy and sorry to bother you again. We wonder if you need any further clarification. We would greatly appreciate it if you'd like to have any discussion with us. Thanks for your time!
>
> Many thanks,
>
> Authors of 1219

---

### Official Review · Reviewer_v6YC · 2022-07-14

**Rating:** 7
**Confidence:** 4
**Soundness:** 3 good
**Presentation:** 3 good
**Contribution:** 3 good

**Summary:**

The paper proposes a mechanism to perform fairness aware feature selection that achieve counterfactual fairness under certain assumptions.




**Questions:**

Please clarify the connection to related work and how sensitive this approach is with respect to noise in MPDAG.

**Limitations:**

The authors have clearly discussed their assumptions.

**Strengths And Weaknesses:**


S1 Fairness aware feature selection is an important problem which is underexplored.
S2 Problem setting that complete causal graph may not be available is important.
S3 The paper is easy to follow.


Weaknesses
W1 The critical assumption is that the sensitive attribute does not have any ancestors. Another crucial assumption for the analysis is the absence of confounders. Please discuss the realistic implications of these assumptions and the challenges because of these.

W2 It is unclear how noise in MPDAG would affect the quality of this approach. Please discuss this in light of cases where it is learned from data.

W3 Experiments on real data do not show how their approach is able to ensure fairness of the trained classifier.

W4 Related Work: There is no discussion of these techniques as compared to this recent paper which does not assume knowledge of causal graph and performs feature selection.
[1] Causal Feature Selection for Algorithmic Fairness. S Galhotra, K Shanmugam, P Sattigeri, KR Varshney. SIGMOD 2022.

W5 Algorithm complexity seems quite high. Please discuss running time on real datasets.

---

> ### Author Response · Authors · 2022-08-02
> **Reply to Reviewer v6YC (2/2)**
>
> > W4 Related Work: There is no discussion of these techniques as compared to this recent paper which does not assume knowledge of causal graph and performs feature selection. [1] Causal Feature Selection for Algorithmic Fairness. S Galhotra, K Shanmugam, P Sattigeri, KR Varshney. SIGMOD 2022.
>
> Thanks for reminding us about this recent work. We found it was published in June 2022, after the NeurIPS submission date. By performing conditional independence tests between different feature subsets in the context of data integration, Galhotra et al. [1] studied the problem of fair feature selection without assuming access to the underlying graph. However, they did so from the perspective of interventional fairness at the subpopulation level as opposed to the individual-level counterfactual fairness in our work. We have added a discussion on this recent paper in the revision according to your kind suggestion.
>
> > W5 Algorithm complexity seems quite high. Please discuss running time on real datasets.
>
> Thanks for raising this concern. In the general case, the complexity in the worst case is $\mathcal{O}(|sib(S,\mathcal{G})+ch(S,\mathcal{G})|\*|E(\mathcal{G})|\*|V(\mathcal{G})|)$ (L237 in the first manuscript), which scales linearly w.r.t edges and nodes.
> Under the root node assumption, the computational complexity further reduces to $\mathcal{O}(|V|+|E|)$ (L266 in the first manuscript), where $|V|$ is the number of nodes and $|E|$ is the number of edges in $\mathcal{G}$. On the real dataset with 30 nodes and 17 edges, our algorithm running time is 0.6499 ms in a Macbook Pro with 2.7 GHz Dual-Core Intel Core i5.
>
> > Questions: Please clarify the connection to related work and how sensitive this approach is with respect to noise in MPDAG.
>
> Thanks for your kind comments. We have clarified the connection to the related work in the Introduction part, specifically on the causal fairness and causal discovery methods on Line 24-48 in the original manuscript. Besides, we have provided a summary of existing results on ancestral relations identifiability in Appendix G and referenced in Line 63 in the main of the original manuscript. As for the sensitivity of this approach with respect to noise in MPDAG, see our response to W2. Please let us know if you have any further concerns, and we are encouraged to have a discussion.

---

> ### Author Response · Authors · 2022-08-02
> **Reply to Reviewer v6YC (1/2)**
>
> Thank you for your constructive comments. We address your comments point by point as follows. We have also tried to revise the manuscript according to your suggestions.
>
> > W1 The critical assumption is that the sensitive attribute does not have any ancestors. Another crucial assumption for the analysis is the absence of confounders. Please discuss the realistic implications of these assumptions and the challenges because of these.
>
> Thanks for this insightful question.
>
> - Response to Assumption on "sensitive attribute does not have any ancestors".  First, we would like to mention that we have provided a general solution without this assumption (section 5.1 General case). In section 5.2, we provided the solution under this assumption, and the justifications can be found in L245-L249 in the original manuscript. To sum up, there are many situations where the protected attribute like gender and age cannot be caused by other features. In this case, we can obtain the same results as if we know the full DAG, which is very interesting and useful in situations where the assumption holds.
>
> - Response to Assumption of no confounders. We have stated the reason (Line 357-359 in the original manuscript) in the Conclusion and discussion part that ``because the causal discovery algorithms themselves will not work well in such challenging scenarios.'' To sum up, causal discovery is a challenging ill-posed problem, and some assumptions, e.g., no confounders, are commonly adopted for the initial development of a practical approach, e.g., PC and GES. It remains an active research area to relax this assumption which has attracted much attention.
>
> - In addition, we have followed the prior work on establishing counterfactual fairness [66, 4, 7 , 57 ] which also assumes no selection bias and confounders. Though our method cannot handle confounders, we are the first to consider the problem of unknown or partially known DAGs (L64-67 in the original manuscript), which makes a steady step toward more practical counterfactual fairness.
>
> - Finally, we have noticed a recent work "Selection, Ignorability and Challenges with Causal Fairness" by Fawkes et al.[1]. According to their discoveries, with the two critical assumptions on a causal DAG, a counterfactual fairness measure degenerates to demographic parity, which is discussed extensively by the authors [1]. In this situation, it is considerably more difficult to provide a clear causal interpretation without knowing the causal DAG. This may give rise to an exciting topic to research in future work. Given the reviewer's concern, we have reflected the current debate surrounding these conditions in the 'Conclusion and discussions' part of the revision.
>
> Hope this addresses your concern and please kindly let us know if you have further concerns.
>
> [1] Fawkes, Jake, Robin Evans, and Dino Sejdinovic. "Selection, Ignorability and Challenges With Causal Fairness." First Conference on Causal Learning and Reasoning. 2021.
>
> > W2 It is unclear how noise in MPDAG would affect the quality of this approach. Please discuss this in light of cases where it is learned from data.
>
> Thanks for raising this concern. At this point, we do not know, theoretically, how noise in MPDAG would affect the performance of our FairRelax when the graph is learned from data. However, we have endeavoured to explore this experimentally in Appendix F.8 in the original manuscript, but we feel sorry for missing the reference in the main paper. After comparing Table 1 and Table 5 in Appendix F.8, we concluded that "there is not much difference on fairness and prediction performance on FairRelax model in two cases". Please refer to Appendix F.8 for the detailed numeric results. Besides, we have also stated in the second footnote in page 7 that "Given a sufficiently large sample size, current causal discovery algorithms can recover the CPDAG with high accuracy on the simulated data [19]." Again, we are sorry for missing the reference to Appendix F.8 and have added it in the revision.
>
> > W3 Experiments on real data do not show how their approach is able to ensure fairness of the trained classifier.
>
> Our approach is the "FairRelax" model. In Line 341 in the original submission, we have stated that "The results are reported in Table 2. Since under the root node assumption, there is no possible descendants of the sensitive attribute, the model "Fair" and "FairRelax" give the same RMSE result and both of them achieve counterfactual fairness at the cost of slight accuracy decrease."

---

> > ### Comment · Reviewer_v6YC · 2022-08-08
> > **Response to authors**
> >
> > Thanks for answering my questions. I appreciate that authors have answered all questions. I am raising my rating to Accept.

---

> > > ### Author Response · Authors · 2022-08-09
> > > **Reply to Reviewer v6YC**
> > >
> > > Thanks for strengthening our paper, and we appreciate your efforts!
> > >
> > > Authors of 1219

---

> ### Author Response · Authors · 2022-08-05
> **Reply to Reviewer v6YC**
>
> Dear Reviewer v6YC,
>
> We appreciate your comments and time! We have provided answers to your questions and revised the paper following your suggestions. Would you mind checking it and confirming if you have further questions?
>
> Best Regards,
>
> Authors of 1219

---

### Review · Ethics_Reviewer_mze4 · 2022-07-31

**Recommendation:** None.

**Ethics Review:**

No ethical concerns.

---

### Review · Ethics_Reviewer_q7ME · 2022-08-01

**Recommendation:**

Given the narrow focus of the work, the ethical issues are ultimately minor, when considered in context.

Nevertheless, the paper would still benefit from a deeper and more honest account of how restrictive the presumed limitations of the proposed method are in practice, and to what extent these obstacles are practically possible to overcome. The paper would also benefit from a deeper engagement with well-known critiques of how counterfactual fairness methods tend to be pitched and used in practice, as given for example in "The Use and Misuse of Counterfactuals in Ethical Machine Learning". I would encourage the authors to extend the discussion of the limitations, as many readers would not have the required level of familiarity, and may erroneously believe that they can use the method in their own applied scenario, even in case when it may not be safely and ethically applicable. To mitigate such risks, some additional details should be provided.

**Ethics Review:**

This paper proposes a method for identifying ancestral relations on MPDAGs, which can be learned from observational data and domain knowledge. This is then exploited for demonstrating good performance in achieving counterfactual fairness, as knowing the causal graph is important for the application of counterfactual fairness methods. This proposed approach is then evaluated on both the synthetic data, as well as the UCI student performance data set. The authors don't discuss how representative this data is, of the types of real-world problems where their method could be applied, nor whether the particular choice of benchmark perpetuates biases in the utilisation of Western data benchmarks in the development of fairness methods. That being said, given the scope of the paper, and the particularities of the method being proposed, it feels like these are meant to be illustrative examples of the method 'working as intended' rather than a deep empirical evaluation of its use across a number of domains. With that in mind, these are minor issues in this particular context.

As for the limitations, as the authors themselves state: "Throughout this paper, we assume no selection bias and presence of confounders  because the causal discovery algorithms themselves will not work well in such challenging scenarios". This is reasonable in-context, as the proposed method has a fairly narrow focus and the authors wanted to highlight its benefits within that use case. However, this is -not- a reasonable assumption overall, for applications on real-world data. The authors do not overclaim and are communicating this limitation to the reader.

Yet, the paper would still benefit from the authors highlighting more just how prevalent these issues are, putting the applicability of the proposed approach in context; as well as discussing at a greater length what they see as the likely / necessary next steps to make such methods safely applicable in practice, in presence of confounders and selection bias.

Given that the authors focus on counterfactual fairness, they should also openly engage with the criticisms presented in "The Use and Misuse of Counterfactuals in Ethical Machine Learning" and flag those risks in the paper.

---

### Author Response · Authors · 2022-08-02
**Response to All Reviewers**

We appreciate all reviewers' work and friendly comments. We are encouraged that they found our paper to be well-motivated (Reviewer Hhvs), important (Reviewer v6YC), useful (Reviewer UUSP), relevant, new, and interesting (Reviewer Hhvs). Moreover, we are grateful that reviewers recognised our contributions both on technical and application parts (Reviewer Hhvs). Reviewers also found that our simulation uses fairly high dimensional DAGs compared to the common examples considered in counterfactual fairness (Reviewer UUSP). We also appreciated that reviewers found our paper is easy to follow (Reviewer v6YC), well-written and well-organized (Reviewer Hhvs).

The modification in the manuscript is summarised as follows:
- We tackle questions on the technical part raised by Reviewer Hhvs.
- We try our best to clarify the technical contributions in a high level in the main paper.
- We modify some introduction and discussion to respond to reviewers' comments and move Algorithm 1 in the original manuscript from the main text to the appendix. Some minor issues are revised accordingly.

We also appreciate reviewers pointing out our weaknesses. We address their comments point by point and try our best to update the manuscript accordingly. The modification part is coloured in blue. Hope our response addresses the reviewers' concerns.

---

### Author Response · Authors · 2022-08-08
**Response to Area Chair**

Dear Area chair,

Thanks for handling our submission. Only one day is left in the Author-Reviewer Discussion period, yet we did not receive any response to our rebuttal from Reviewer v6YC or Reviewer UUSP. We understand that reviewers are very busy. We wonder if they need further clarification. We would greatly appreciate it if you could follow up on this issue.

Many thanks,

Authors of 1219

---

### Author Response · Authors · 2022-08-10
**Can you please help discuss the following in the AC-Reviewer discussion phase？**

Dear AC and reviewers,

Thank you very much for reviewing our paper. After several rounds of discussions, there is one remaining concern from reviewer UUSP. We briefly summarize the concern and our response below.

Reviewer UUSP's concern is mainly about the practicability of using non-descendants of sensitive attributes for fair prediction. The concern is based on the following two points, which are not true according to our understanding.

First, reviewer UUSP believes that sensitive attributes should causally influence every variable of interest, such that it is impossible to find non-descendants. However, reviewer UUSP did not provide strong scientific support for this claim. The provided wiki page is not convincing, and the page is only about “race”. Moreover, in our real data experiments, we have clearly shown that only 4 out of 30 features are descendants of the sensitive attribute “sex”.

Second, reviewer UUSP misunderstands the seminal counterfactual fairness paper [Kusner et al., 2017]. Reviewer UUSP does not believe the [Kusner et al., 2017] paper intended to establish its Lemma 1 as a standard or default approach and thinks it encourages to use descendants. However, the fact is that Lemma 1 in [Kusner et al., 2017] is indeed the standard of their methods. All three levels of assumptions in [Kusner et al., 2017] are based on its Lemma 1. Aside from the level 1 which our work builds on, two other levels of assumptions cannot depend on descendants as well, which is discussed in section 4.2 and illustrated explicitly in Figure 2. The latent variable in level 2 and the error term in level 3 used for prediction are both non-descendants of the sensitive attributes.

Besides, it is obvious that the authors of [Kusner et al., 2017] do not encourage to use descendants in prediction, as the explanation of Lemma 1 in section 3.2,
> This does not exclude using a descendant W of A as a possible input to $\hat{Y}$. **However, this will only be possible in the case where the overall dependence of $\hat{Y}$ on A disappears, which will not happen in general.** Hence, Lemma 1 provides the most straightforward way to achieve counterfactual fairness.

Note that **"which will not happen in general"** is contradictory to reviewer UUPS’s viewpoint, as reviewer UUSP thinks the disappearance of the overall dependence of $\hat{Y}$ on A (cancel out) is common. While this might be a point worth more debating, it is not the focus of our paper. We do not think the possible debate should hurt our contribution in any way.

We sincerely hope you could discuss on this point during the AC-Reviewer discussion phase.

Many thanks,

Authors of paper 1219

---

> ### Comment · Reviewer_UUSP · 2022-08-10
> **More concise version of my other reply**
>
> I believe the value in the current paper is mainly as a fairly general SCM discovery algorithm. The algorithm relies on an ancestral closure property which some causal fairness methods assume. Hence the paper motivates this algorithm as an application to counterfactual fairness, as indicated in the title of the paper. I believe this is a poor motivation/application, because the version of fairness it focuses on is too narrow and unrealistic, namely the sufficient--but not necessary--condition of Lemma 1 in [Kusner et al., 2017] regarding using only non-descendants of the sensitive attributes. I believe the paper would be stronger if it were framed as a more general purpose causal discovery algorithm and not specifically as a method for counterfactual fairness.
>
> There was much discussion about my contention that the [Kusner et al., 2017] Lemma 1 version of fairness is too narrow and unrealistic. I have replied elsewhere (in the thread of responses to my review) in more detail to the authors most recent points about this above. The most important takeaways here should be that (1) the real data experiments cannot be used as a proof because we do not know a ground truth for comparison, and (2) the authors are incorrect about the Levels 2 and 3 in [Kusner et al., 2017], for example as that paper mentions in an application:
>
> > We estimate the error terms by first fitting two models that each **use race and sex** to individually predict GPA and LSAT. We then compute the residuals ... We use these residual estimates ... to predict FYA. We call this Fair Add
>
> I maintain that only using non-descendants of sensitive attributes is a narrow and unrealistic version of fairness, and hence that it is a poor (and unnecessary) choice for the title and main framing of the current paper.

---

> > ### Author Response · Authors · 2022-08-10
> > **Further Reply**
> >
> > 1. Please notice that our FairRelax method also uses possible descendants to make prediction.
> >
> > 2. We believe we are correct about the Levels 2 and 3 in [Kusner et al., 2017].
> > > We estimate the error terms by first fitting two models that each use race and sex to individually predict GPA and LSAT. We then compute the residuals ... We use these residual estimates ... to predict FYA. We call this Fair Add.
> >
> >     Please notice that the error terms in level 3 [Kusner et al., 2017] used to make prediction are **non-descendants and also independent of the race and sex** , as mentioned in section 5,
> >     > In Level 3, we model GPA, LSAT, and FYA as continuous variables with additive error terms **independent of race and sex** (that may in turn be correlated with one-another).
> >
> > 3. > the real data experiments cannot be used as a proof because we do not know a ground truth for comparison.
> >
> >      Since our paper is not intended to test the causal discovery algorithm, we do not need the ground-truth causal graph. Besides, the paper [Kusner et al., 2017] is based on an assumed causal graph, without the ground-truth graph as well. If the causal discovery algorithm is trustable, our method is valid in practice.

---

> > > ### Comment · Reviewer_UUSP · 2022-08-10
> > > **More misunderstanding**
> > >
> > > Error terms (or latent variables) are never observed, they are estimated. The level 2+ methods do that by explicitly using the sensitive attributes (and, depending on the structure, their descendants). Hence the actual function that computes predictions explicitly uses the sensitive attributes (and etc). You can see this in the math formulas in that paper which I replaced by "..." in my quote. The function explicitly takes R and G as inputs.
> > >
> > > Your argument from the real data experiment was that it showed an example where some variables are not descendants of the sensitive attribute. You cannot reach that conclusion based on the output of your algorithm alone, because, as you know, the algorithm requires additional modeling assumptions (e.g. no unmeasured confounding) for its output to have any guarantee.

---

> > ### Author Response · Authors · 2022-08-10
> > **An extremely critical misunderstanding**
> >
> > > I believe the value in the current paper is mainly as a fairly general SCM discovery algorithm. The algorithm relies on an ancestral closure property which some causal fairness methods assume. Hence the paper motivates this algorithm as an application to counterfactual fairness, as indicated in the title of the paper. I believe this is a poor motivation/application, because the version of fairness it focuses on is too narrow and unrealistic,…
> >
> > **Please note that our method is not a SEM discovery method. Instead, we rely on the output of the current causal discovery methods to identify ancestor relations.** I do not think we need to answer your following points given that your first point is already wrong.

---

> > > ### Comment · Reviewer_UUSP · 2022-08-10
> > > **It is only a name**
> > >
> > > I said "SCM discovery" merely as the general category of methods I would describe this fitting under, with identifying ancestor relations as a special case enabled by some additional assumptions (e.g. the ancestral closure assumption of a sensitive attribute)
> > >
> > > Please do not make the discussion tedious by objecting over terminology. I have already dedicated more time to this discussion than I believe is usually expected of reviewers, so I prefer to keep it focused and reach a conclusion.

---

### Meta-Review · Area_Chair_Wvcz · 2022-09-01

**Recommendation:** Accept
**Confidence:** Less certain

**Metareview:**

This paper has divergent views in the sense two reviewers have given positive assessments (6 and 7) while the other reviewer has given a negative assessment (score of 3). This paper also had very 'heavy' discussions between the reviewer with negative opinion and the authors.

First of all I would like to thank the reviewer involved in patiently discussing with the authors dedicating valuable personal time.
Let me start with the aspects all reviewers *more or less agree* on :

a) The main technical piece is an efficient algorithm and provable guarantees for identifying definite non-descendants and definite descendants from an MPDAG - maximum partially directed acyclic graph - the equivalence class of Causal DAGs one obtains after incorporating any arbitrary side information. Previous such results were known for CPDAGs and they don't carry over to MPDAGs. Therefore it is a non trivial result (specifically Lemma 4.4 ). So all reviewers agree that finding definite non descendants in Equivalence classes that also include side information is a very solid contribution.


b) The aspect in which reviewers had divergent opinion is this:  the paper's claim to be able to train counterfactual fair classifiers leveraging the result from [Kusner et. al 2017] that any function of non-descendants is counterfactually fair.

One of the reviewer's strong contention is that in most fairness datasets, most variables that are highly predictive of outcomes will also be downstream of sensitive attributes like race etc.. and therefore relying only on non-descendants is not exactly a realistic application. Authors cited their empirical structure learning results that show very few descendants and comments from Kusner et. al 2017 paper to bolster their case. Reviewer responded by citing alternate statements from the same paper etc..

*My opinion* is that in a specific context when fairness with respect to a specific sensitive attribute is desired, there are also often other features that has no causal relationship with the sensitive attribute but has a *correlation* (Examples include age and race, race and gender etc.. ).  To cite a recent reference please see Example 15 in https://arxiv.org/pdf/2207.11385.pdf (this reference is recent and I am *not* expecting authors or anyone else to have known this - it is just to demonstrate the point). The example shows *testable* correlations between sensitive attributes and non-descendants in COMPAS and Adult datasets.

This shows that a) neither causal sufficiency  nor b) the non-existence of non-descendants are realistic . In fact, spuriously related non-descendants give rise to spurious bias which may not be an object of correction for fairness (broadly speaking). This shows that causal sufficiency is a strong assumption (as authors have assumed) and also non-descendants do exist.

c) Another point to be noted is that Kusner et. al. 2017 do consider confounded models unlike the authors. Once you view exogenous, endogenous (observed) variables and sensitive attribute as one full deterministic system, their point is ALL exogenous + non descendant endogenous variables are "non-descendants" topologically and therefore could be used. They did not imply non-descendants endogenous 'only' as the authors contend in their discussions.  In fact, the algorithm section in Kusner et. al. 2017 - advocates for sampling Exogenous from some side information (level 2 and 3 information) and forming a predictor as a function of exogenous *and* non descendant endogenous variables.
Therefore, reviewer has a valid point on the discussed aspect as well. Authors may want to pay attention to this.


*In summary*: Authors' contention that Kusner et al 2017 paper advocates for non-descendant endogenous as their main sufficient criterion appears to be not exactly correct. However, non-descendants and their confounding with sensitive attribute is a more realistic model. However, authors core technical structure learning contribution is also noteworthy.

 If this line of work is to be pursued where one could find non descendants even under limited confounding (between sensitive attributes and non- descendants - a mild violation of causal sufficiency) - it would be a step towards obtaining counterfactually fair classifiers (although even such a classifier would have to sacrifice a lot on accuracy depending on how many descendants one observes).  However, even positive reviewers have opined that main strength of the paper is a solid structure learning result that identifies non-descendants in a fully observational setting.

*Recommendation*: In the spirit of not blocking valid ideas that are fundamental and also the fact that one cannot always make the weakest set of assumptions to make progress, I tend to favor acceptance. A *very strong* suggestion to authors - I would place structure learning as the centerpiece and motivate it by a need to learn non-descendants (in the general sense) motivated by Kusner et al 2017. Authors also need to highlight Fair relax - a relaxation that they have proposed that uses possible descendants and definite non descendants to predict - it seems to be closer than other approaches to counterfactually fair one and therefore removing the singular focus on (as the discussions would have one believe) only observed definite non-descendants.




**Award:**

No

---

### Decision · Program_Chairs · 2022-09-14

Accept